# A Unified Perspective and Review on Tree Search for LLMs Test-Time Scaling

## Abstract

As the scaling of large language models (LLMs) during training reaches diminishing returns due to increased resource requirements and limited data availability, focus has shifted toward scalable test-time algorithms. Chain-of-Thought (CoT) reasoning, which enables intermediate reasoning steps in text space, has emerged as a promising approach. However, CoT's **single-path exploration** is susceptible to biases and underexploration of the solution space in complex problems. This survey examines advancements in tree search-based methods for enhancing LLM test-time reasoning. Beginning with foundational search algorithms like depth-first search (DFS) and breadth-first search (BFS), we trace the evolution to heuristic-guided approaches and ultimately Monte Carlo Tree Search (MCTS). We introduce **a unified framework** for comparing these methods, focusing on their core designs, reasoning reward formulations, and targeted applications. Our analysis highlights MCTS's capability to balance exploration and exploitation, overcoming limitations of traditional inference methods like beam search. This survey establishes a foundation for advancing scalable test-time reasoning in LLMs, with implications for improving general-purpose reasoning capabilities.

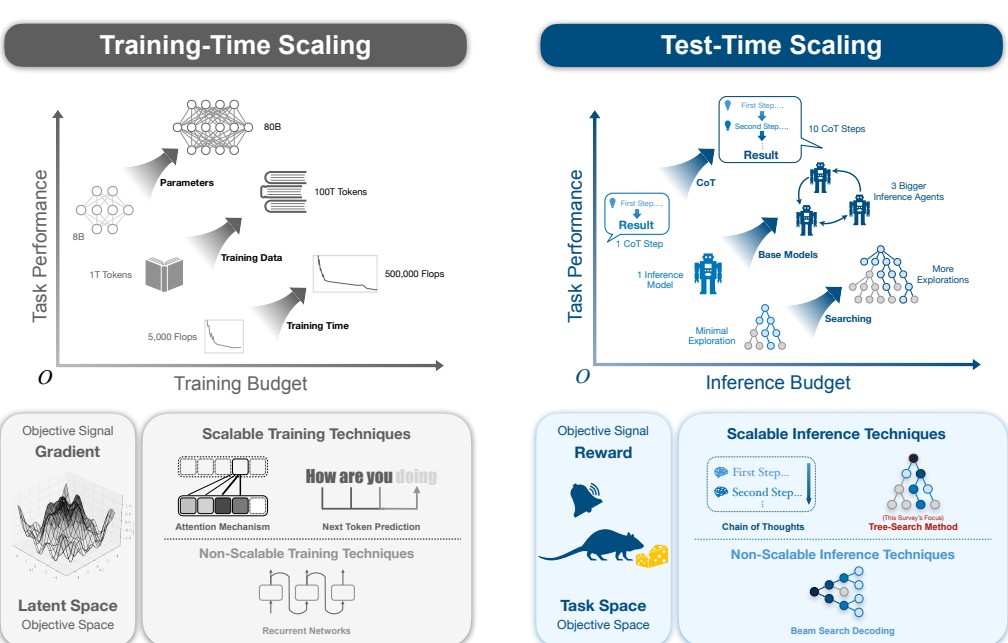

Figure 1: Comparison of Training-time and Test-time Scaling: Highlighting Budget Allocation, Techniques, and Their Impact on Task Performance.

# 1 INTRODUCTION

As returns from scaling Large Language Models (LLMs) during pretraining diminish (Kaplan et al., 2020a; Hoffmann et al., 2022a), the research focus is shifting towards eliciting their full potential by allocating additional computation at inference. This paradigm, known as test-time scaling (*TTS*) (Brown et al., 2024; Wu et al., 2024), is inspired by human cognition, where deeper, more deliberate thinking often yields superior outcomes (Kahneman, 2011; Evans, 1984). Unlike specialized models, general-purpose LLMs leverage natural language to reason within the text space, unlocking a path toward artificial general intelligence (Bubeck et al., 2023). However, conventional search algorithms like beam search have proven inadequate for navigating the vast and complex reasoning spaces required for challenging tasks.

The introduction of Chain-of-Thought (CoT) prompting was a breakthrough, demonstrating that LLMs could externalize latent reasoning processes into explicit, sequential steps (Wei et al., 2022). CoT's performance was shown to scale with the test-time compute budget, yet its reliance on a single, greedy reasoning path remains a significant limitation. To overcome this, recent work has focused on tree search algorithms that explore multiple reasoning paths in parallel (Wang et al., 2023b). These methods, drawing inspiration from classical AI, aim to efficiently traverse a large reasoning tree to discover an optimal solution path, significantly boosting performance on complex reasoning tasks (HuggingFace, 2025).

The evolution of these search algorithms mirrors the progression of classical search: from early uninformed methods analogous to Depth-First and Breadth-First Search (e.g., Tree-of-Thought), to heuristic-guided search, and now to sophisticated strategies like Monte Carlo Tree Search (MCTS). Despite this rapid innovation, the field has become fragmented, characterized by inconsistent notation, varying evaluation protocols, and a lack of a unified conceptual framework. This disorganization hinders systematic comparison and impedes progress.

This survey aims to unify the rapidly growing space of search-based reasoning in LLMs by providing a coherent framework and cross-paper synthesis that extends beyond descriptive summarization. We consolidate the field's core algorithmic designs, surface common structural principles, analyze empirical trends, and identify gaps that are not evident from individual works. **Our main contributions are:**

- **Unified Formalism:** We introduce a common formalism that decomposes tree-search methods into shared components (node representation, evaluation, backup dynamics), enabling consistent comparison and clarifying the role of reward in test-time search.
- **Principled Taxonomy and Cross-Paper Insights:** We organize existing methods along core axes—search mechanism, evaluation signal, and domain structure—and synthesize cross-paper evidence on problem suitability, compute–accuracy trade-offs, and the differences between MCTS and heuristic tree search, while highlighting methodological gaps in evaluation practices.
- **Applications and Future Directions:** We summarize the main applications of search-based reasoning (performance boosting, data generation, distillation) and outline future opportunities in adaptive search and scalable, high-fidelity reward modeling.

# 2 SEARCH IN GENERAL AI

Reasoning tasks can be modeled as a search on a tree or graph, where states branch into subsequent possibilities. The large branching factors in reasoning create a massive search tree, making efficient exploration crucial. AI search algorithms systematically navigate this solution space to find optimal paths by balancing computational cost and accuracy. This section reviews foundational tree-search methods, setting the stage for advanced LLM-based reasoning search. More details could be found in Appendix B.

## 2.1 UNINFORMED SEARCH

**Uninformed search** algorithms like Breadth-First Search (BFS), Depth-First Search (DFS), and Uniform Cost Search (UCS) operate without any knowledge of the goal's location. They rely solely

on the problem's structure-actions, costs, and goal conditions-to explore the search space. Their exploration strategies differ: BFS guarantees finding the shortest path in terms of steps, while UCS finds the lowest-cost path. These methods are systematic but can be inefficient in large state spaces due to their lack of guidance.

## 2.2 INFORMED SEARCH

**Informed search**, or **heuristic search**, uses a domain-specific **heuristic** function, $h(n)$, to guide exploration by estimating the cost to a goal.

$$h(n) = \text{estimated cost of the cheapest path from node } n \text{ to a goal} \tag{1}$$

An *admissible* heuristic never overestimates the true cost, while a *consistent* one satisfies the triangle inequality, $h(n) \leq c(n, n') + h(n')$, where $c(n, n)$ is is the actual cost of going from $n$ to $n'$. Overestimation is disallowed because it can cause the search to overlook the optimal path, whereas underestimation only affects efficiency, not correctness. A more *informed* heuristic (i.e., a tighter lower bound on the true cost) generally leads to more efficient search. Algorithms like A* Search and Beam Search use heuristics to prioritize promising paths. Notably, A* search is guaranteed to find the optimal solution if its heuristic is admissible. The effectiveness of informed search hinges on the quality of the heuristic, balancing its computational cost against the search efficiency it provides.

## 2.3 MONTE CARLO TREE SEARCH (MCTS)

Monte Carlo Tree Search (MCTS) is a statistical search algorithm adapted from two-player games for single-agent LLM reasoning tasks. It excels in large search spaces by balancing exploration and exploitation without a predefined heuristic. MCTS operates in four phases: selection, expansion, simulation, and backpropagation. In the selection phase, it traverses the tree using the *Upper Confidence bounds for Trees (UCT)* policy:

$$a^* = \arg\max_{a \in A(s)} \left[ Q(s, a) + c\sqrt{\frac{\ln N(s)}{N(s, a)}} \right] \tag{2}$$

where $Q(s, a)$ is the estimated value of action $a$, $N(s)$ and $N(s, a)$ are visit counts such that $N(s)$ is the total number of times state $s$ has been visited, and $N(s, a)$ is the number of times action $a$ has been taken from $s$, and $c$ is an exploration constant. Unlike uninformed methods, MCTS uses statistical sampling to handle vast search spaces. Unlike informed methods like A*, it learns its own value function through simulated rollouts, making it highly effective for complex LLM tasks where designing a good heuristic is challenging.

## 2.4 REWARD AS A GUIDING SIGNAL: SEARCH VS. RL

The notion of a "reward" plays a central role in both search and Reinforcement Learning (RL), yet its function and implementation differ fundamentally, as illustrated in Figure 2. Although both search rewards and RL rewards are commonly referred to simply as "reward" in the MCTS and RL literature, they serve distinct purposes. This ambiguity can obscure the conceptual relationship between test-time planning and training-time optimization. Clarifying this distinction is essential for establishing the unified framework developed throughout this paper. Additional details are provided in Appendix D.

**In Reinforcement Learning**, a reward signal is **assimilated into the model's parameters** via gradient-based updates. This process induces a durable shift in the model's underlying policy ($\pi_\theta$), making RL suitable for learning **generalizable, reusable skills**. The reward serves to optimize a universal policy that is expected to perform well across a distribution of related tasks.

**In Test-Time Search**, a reward is an **external, transient signal** used to guide the planning process for a single problem instance. This signal, often from a non-differentiable oracle (e.g., a verifier, a code execution environment), directs the search toward a high-quality solution for the current query. Crucially, it does **not** alter the model's parameters, leaving its general capabilities intact. This makes search ideal for **task-specific, on-the-fly optimization** without risking catastrophic forgetting or policy degradation. RL, by contrast, is vulnerable to catastrophic forgetting because parameter updates for new tasks can overwrite knowledge from prior tasks.

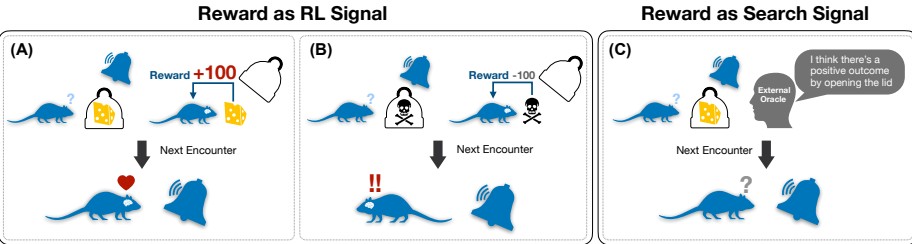

Figure 2: Reward Design: Search vs. RL. (A) In RL, a positive reward updates the agent's policy, making it more likely to repeat the action. (B) A negative reward also updates the policy, discouraging the behavior. The change is **durable**. (C) In search, an external oracle provides a reward signal to guide the current decision process without altering the agent's underlying parameters.

In summary, RL uses rewards for long-term *policy optimization*, whereas search employs them for immediate, instance-level *planning and guidance*. A search reward defines a local, task-specific objective used solely within a single inference-time planning instance. In contrast, an RL reward serves as a global training signal that reshapes the model's parameters over many episodes. Understanding this difference is crucial for interpreting the hybrid MCTS–training approaches, where test-time search signals are leveraged as training data for long-term model improvement.

## 3 MONTE CARLO TREE SEARCH (MCTS) FOR LLMS

### 3.1 UNIFIED PROBLEM FORMULATION AND NOTATION

To provide a clear comparative framework for MCTS-based LLM reasoning, we adopt a unified notation for consistency across methods. **Note:** as in recent LLM planning work, the "environment" is simply the evolving text trace, and transitions are deterministic: each action $a_i$ (a reasoning step) uniquely yields the next state $s_{i+1}$. This is a planning—not stochastic MDP—formulation used in RAP (Hao et al., 2023a), ReST-MCTS (Zhang et al., 2024a), AlphaLLM (Tian et al., 2024a), rStar-Math (Guan et al., 2025), and LLaMA-Berry (Zhang et al., 2025b).

Importantly, states are partial reasoning traces while actions represent only the next incremental step; the two spaces are therefore not equivalent. This asymmetry is intrinsic to deterministic planning and contrasts with RL's environment-driven MDPs. The objective is to find an optimal reasoning trace $p' = [s_1, \ldots, s_n]$ for a problem $Q$. This formulation enables us to unify insights across papers and surface shared structural principles (e.g., how node granularity interacts with evaluation), which prior works have discussed only in isolation.

Table 1: Unified Notations for MCTS-Based Methods in LLM.

| Symbol | Definition |
|---|---|
| $Q, c$ | Problem question and conditioning prompt |
| $s_i, a_i$ | Reasoning state and action at step $i$ |
| $p_i$ | Partial reasoning trace $[s_1, s_2, \ldots, s_i]$ |
| $v_i, r_{s_i}$ | Value of trace $p_i$ and reward for state $s_i$ |
| $\pi, V_\theta, R_\theta$ | Policy (LLM), value, and reward models |
| $T_Q, \mathcal{A}$ | Search tree for problem $Q$ and the action space |
| $C_i = (t_i, n_i, q_i)$ | Tree node with identifier $t_i$, visit count $n_i$, and quality value $q_i$ |

### 3.2 STRUCTURING THE SEARCH: NODE REPRESENTATION AND GRANULARITY

A fundamental design choice is the definition of a node in the search tree $T_Q$, which dictates the granularity of the search. We identify three primary strategies:

**Trace-based nodes**, employed in step-driven frameworks like ReST-MCTS* (Zhang et al., 2024a), define each node as a complete partial reasoning trace $p_i = [s_1, \ldots, s_i]$. This representation allows the value function $v_i = V_\theta(p_i)$ to capture the full context of the preceding reasoning path when assessing a node's potential.

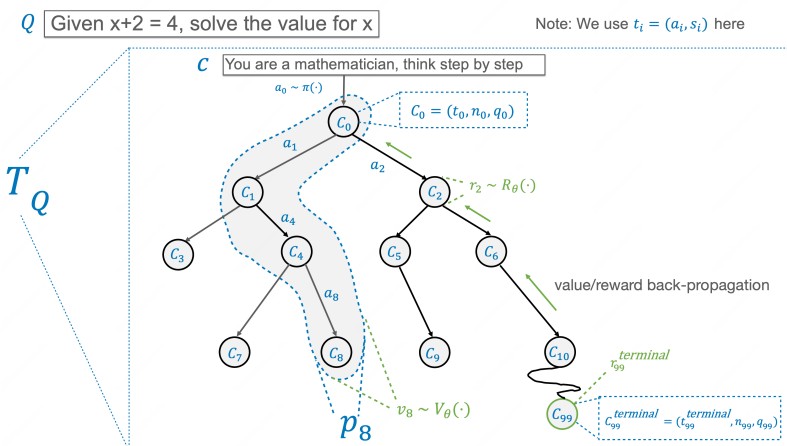

Figure 3: Unified Notations for MCTS-Based Methods in LLM.

**State-Action nodes**, used in methods such as RAP (Hao et al., 2023b) and ALPHALLM (Tian et al., 2024b), represent each node as a state-action pair $(s_i, a_i)$. This more localized view focuses evaluation on the immediate quality of a single reasoning step, simplifying the input to the reward model.

**Terminal-State nodes**, a hallmark of purely goal-driven approaches like LLaMA-Berry (Zhang et al., 2024d) and MCTSr (Zhang et al., 2024c), radically restructure the search space. Here, each node represents a complete, terminal solution $s^{\text{terminal}}$. The tree does not model the sequential generation of a single solution but rather a space of candidate solutions, where edges correspond to refinement or rewriting operations. This transforms the problem from finding an optimal path to finding an optimal node.

In practice, these node definitions correspond to different textual granularities: trace-based nodes typically bundle multiple sentences or "reasoning steps", state-action nodes can align with a single reasoning step or short segment, and terminal-state nodes treat entire solutions as atomic. Finer granularity provides more flexible guidance but increases branching and evaluator cost, while coarser granularity reduces tree size at the cost of less precise feedback.

### 3.3 THE CORE CHALLENGE: DESIGNING THE EVALUATION FUNCTION

The most critical differentiator among MCTS-based methods is the design of the evaluation function, which assigns a quality score ($v_i$ estimate how likely to get to correct solution from current node and $r_i$ estimates how much reward for single step action to current node ) to the given node. This function steers the entire search process, and its design reflects the overarching strategy of the framework.

#### 3.3.1 EVALUATION LOCUS: PROCESS VS. OUTCOME REWARDS

The evaluation signal can be derived from the quality of the reasoning process itself or from the final outcome. Methods focused on improving the reasoning trace, such as ReST-MCTS*, employ a **Process Reward Model (PRM)** or a value function $V_\theta(p_i)$ that evaluates intermediate, non-terminal states. This provides fine-grained, step-by-step guidance, encouraging the discovery of high-quality reasoning paths that can be used for subsequent model training.

Conversely, methods that prioritize finding the correct final answer often rely on an **Outcome Reward Model (ORM)**. In this paradigm, intermediate nodes receive a default reward (e.g., 0), and a significant reward is assigned only to terminal nodes $s^{\text{terminal}}$. This terminal reward can be determined by various means: majority voting as in rStar (Qi et al., 2024), execution against test cases as in PG-TD (Zhang et al., 2023a) and RethinkMCTS (Li et al., 2024b), or evaluation by another LLM as in MCTSr and TS-LLM (Feng et al., 2023). HiAR-ICL offers a hybrid approach, capable of operating with either a PRM, an ORM, demonstrating the flexibility of these evaluation schemes.

### 3.3.2 EVALUATOR ARCHITECTURE: EXTERNAL MODELS VS. SELF-EVALUATION

The mechanism for generating rewards is another key design axis. Many advanced systems train a separate, dedicated model for evaluation. For instance, ReST-MCTS* and TS-LLM fine-tune specialized value ($V_\theta$) and reward ($R_\theta$) models on datasets of reasoning traces, learning to predict a trace's potential or a solution's correctness. LLaMA-Berry introduces a Pairwise Preference Reward Model (PPRM), fine-tuning a small LLM to rank solutions against each other.

An alternative, more resource-efficient strategy is to repurpose the same LLM used for policy generation ($\pi$) to also serve as the evaluator. RAP exemplifies this by using its LLM as a "world model" to predict not only the next state but also an associated reward $r_i$. Similarly, MCTSr uses the base LLM to assign scores to solutions through a robust resampling process. This self-evaluation approach reduces the need for external training data and separate model maintenance.

### 3.3.3 MULTI-CRITIC AND COMPOSITE REWARD FUNCTIONS

To capture a more holistic view of node quality, some frameworks combine multiple evaluation signals. ALPHALLM implements a sophisticated multi-critic approach where the node value $Q_i$ is a weighted sum of signals from a value model, a PRM, and an ORM:

$$Q_i \leftarrow \beta_{\text{v}} V(p_i) + \beta_{\text{PRM}} R_{\text{PRM}}(s_i) + \beta_{\text{ORM}} \mathbb{E}[R_{\text{ORM}}(s^{\text{terminal}})]$$

This allows the search to balance long-term potential, immediate step quality, and final outcome correctness. Similarly, RethinkMCTS combines execution-based rewards from test cases with an LLM's self-evaluation score for solutions that pass all public tests, adding a layer of semantic assessment beyond functional correctness. LLaMA-Berry also computes a composite score by blending a "local" reward (comparison to adjacent solutions) with a "global" reward derived from a win-loss matrix against all other explored solutions.

## 3.4 ADAPTING THE MCTS ALGORITHM

Beyond evaluation, methods often introduce custom modifications to the classic MCTS selection, expansion, and backpropagation phases to better suit the domain of LLM reasoning.

In the **selection** phase, many methods like PG-TD and rStar augment the standard UCB1 formula with policy network priors, creating a P-UCB (Polynomial Upper Confidence Bound for Trees) variant. This helps prioritize nodes that are not only promising according to search history but also likely under the base LLM's policy.

The **expansion** phase is also a site of innovation. LLaMA-Berry incorporates a "critique-and-rewrite" step during expansion, where new nodes are generated by refining existing solutions. RethinkMCTS introduces a "rethink" operation for nodes that fail test cases, using verbal feedback to guide the correction of erroneous reasoning steps.

Finally, the **backpropagation** of value updates is tailored to the specific reward structure. While many methods use standard averaging or maximization (e.g., $Q_i \leftarrow \max_{j \in \text{Children}} Q_j$), others propose unique update rules. MCTSr, for instance, updates a parent's value by averaging its current value with the maximum value among its children, providing a smoother value progression: $Q'_i \leftarrow \frac{1}{2}(Q_i + \max_j Q_j)$. This diversity in algorithmic adaptation highlights the flexibility of the MCTS framework in addressing the unique challenges of generative reasoning tasks.

## 3.5 ADVANCED TOPICS AND HYBRID APPROACHES

As the field matures, researchers are exploring more sophisticated techniques that refine the core search paradigm, create better reward signals, and combine multiple methodologies. One prominent direction involves **multi-agent and collaborative search**, moving beyond the single-agent paradigm. Instead of one LLM performing a search, these approaches employ multiple agents that collaborate, debate, or take on specialized roles to solve problems more effectively (Gan et al., 2025; Li et al., 2025b; Yang et al., 2025b; Hou et al., 2025). This collaborative model leverages diverse reasoning pathways and collective expertise to tackle complex challenges like software issue resolution and hierarchical task orchestration, mitigating the limitations of a monolithic agent.

Further advancements focus on the core components of the search process itself, particularly in **reward model design and optimization** and **search efficiency**. The success of any search algorithm hinges on its reward function, and the field is moving towards more granular process-supervised reward models (PRMs) that provide step-level feedback, rather than relying solely on final outcomes Yu et al. (2023a); Ma et al. (2023). To create this fine-grained preference data scalably, many works now employ MCTS to autonomously generate step-level supervision for training robust reward models, reducing the need for costly manual annotation (Luo et al., 2024; Ma et al., 2025; Jin et al., 2025; Brandfonbrener et al., 2024a; Wu et al., 2023). Concurrently, researchers are tackling the high computational cost of tree search. Efforts to improve efficiency include developing more adaptive and intelligent dynamics, such as information-directed search to prioritize valuable feedback (Chandak et al.), dynamic node selection (Wang et al., 2024a; Asai, 2025), and dynamic abstraction dropping to manage complexity (Schmöcker et al., 2025). Other strategies boost performance through improved single-step reasoning (Zhang et al., 2025a) or by making the architecture itself more dynamic, such as with test-time depth adaptation of model layers (Li et al., 2025e), all contributing to a more powerful and efficient search process (Agarwal et al., 2025).

## 3.6 APPLICATIONS OF MCTS

This section offers a concise, practitioner-oriented guide to choosing effective MCTS configurations for major LLM task domains. Each subsection links applications—such as reasoning, code generation, agentic tasks, RAG, and self-improvement—to suitable patterns of node representation, reward modeling, and evaluation. These mappings, together with the algorithmic instantiations in Appendix E, help practitioners quickly identify components appropriate for their use cases.

### 3.6.1 MCTS FOR DIRECT TEST-TIME ENHANCEMENT

This category covers methods that apply MCTS at inference to refine an LLM's response without altering model weights. Instead of relying on greedy or beam search, these approaches explore a tree of reasoning paths or generation steps, guided by value or reward signals, to identify higher-quality outputs. The tradeoff is extra computation at runtime in exchange for greater accuracy, coherence, or adherence to constraints—particularly useful for tasks where the most probable initial trajectory is suboptimal.

A significant body of work focuses on creating domain-agnostic enhancements to MCTS for LLMs, aiming to improve general reasoning and problem-solving capabilities. These studies concentrate on challenges such as search efficiency, interpretability of the reasoning process, and the overall quality of the generated thoughts or solutions (Chen et al., 2024e; Gao et al., 2024; Hui et al., 2024; Wang et al., 2024a; Kang et al., 2024; Zhao et al., 2024; Ding et al., 2023; Pan et al., 2025a). Mathematical reasoning has become a particularly popular domain for applying MCTS. The discrete nature of mathematical problems and the existence of clear, verifiable solutions make it an ideal testbed for defining robust reward functions, which are crucial for guiding the search process effectively toward a correct final answer (Zhang et al., 2024c; Xu, 2023; Yang et al., 2024; Yu et al., 2023a; Zhang et al., 2025c; Luo et al., 2024; Lin et al., 2025b). Similarly, in code generation and software engineering, MCTS is employed to navigate the vast combinatorial space of possible code implementations. The search is often guided by explicit feedback from compilers, unit tests, or formal verifiers, allowing the model to explore, backtrack, and refine code snippets to meet functional requirements (DeLorenzo et al., 2024; Li et al., 2024b; Brandfonbrener et al., 2024b;a; Dainese et al., 2024; Wang et al., 2024d; Zhang et al., 2024e; Xu et al., 2024a; Antoniades et al., 2024; Li et al., 2025b; Wang et al., 2025c; Hu et al., 2025a).

MCTS also provides a principled planning mechanism for LLM agents operating in interactive environments, where an agent must execute a sequence of decisions to achieve a specific goal. In these scenarios, MCTS allows the agent to simulate and evaluate possible action sequences, balancing exploration of new strategies with exploitation of known successful paths (Koh et al., 2024; Zhao et al., 2023; Li et al., 2024c; Murthy et al., 2023; Chi et al., 2024; Zhou et al., 2023; Zhang et al., 2025f; Yu et al., 2023b; Gan et al., 2025; Lin et al., 2025a; Gao et al., 2025; Xie et al., 2025; Li et al., 2025d; Hou et al., 2025). In the context of Retrieval-Augmented Generation (RAG) and other knowledge-intensive tasks, MCTS helps the model strategically decide when to query an external knowledge source and what information to retrieve. This integration of planning with retrieval allows the LLM to dynamically augment its internal knowledge with relevant, up-to-date facts, thereby improving

the accuracy and factuality of its outputs (Wu et al., 2023; Xu et al., 2024b; Jiang et al., 2024; Tran et al., 2024; Wang et al., 2024i; Huang et al., 2024; Choi et al., 2023; Feng et al., 2025a; Luo et al., 2025; Xiong et al., 2025; Dou et al., 2025; Hu et al., 2025b; Gu et al., 2025; Kim & Kim, 2025). Furthermore, MCTS is being explored in the nascent field of multimodal reasoning. For tasks that involve processing and reasoning over both text and images or videos, the search algorithm can explore different strategies for grounding textual logic in visual information, effectively bridging the gap between different modalities to arrive at a coherent and contextually accurate conclusion (Yao et al., 2024a; Dong et al., 2024; Wu et al., 2025a; Yang et al., 2025a).

### 3.6.2 MCTS FOR SELF-IMPROVEMENT VIA DATA GENERATION

A powerful paradigm uses Monte Carlo Tree Search (MCTS) not merely to find a single optimal answer, but to generate extensive sets of high-quality reasoning trajectories. These trajectories serve as synthetic data to fine-tune either the Large Language Model (LLM) itself or an associated reward model, establishing a virtuous cycle of self-improvement. This approach is heavily inspired by seminal concepts in reinforcement learning, such as the self-play mechanism of AlphaZero and preference optimization techniques like Direct Preference Optimization (DPO). Foundational frameworks have demonstrated how to integrate MCTS into a self-training loop, using process rewards and iterative preference learning to progressively enhance the model's reasoning capabilities. These core methodologies enable the LLM to autonomously generate its own training data, refining its policy and value functions through repeated exploration and exploitation of the reasoning space (Guan et al., 2025; Feng et al., 2023; Wang et al., 2024g; Tian et al., 2024b; Xie et al., 2024b; Putta et al., 2024; Qi et al., 2024; Chen et al., 2024a; Wang et al., 2024f; Chen et al., 2024b; Ding et al., 2025; Yuan et al., 2025; Shi et al., 2025b; Kim et al., 2025; Wang et al., 2025c).

The self-improvement paradigm has been extended beyond general reasoning to a wide array of specialized domains. In the context of general LLM capabilities and alignment, MCTS-driven data generation has been employed for sophisticated instruction tuning, automated prompt optimization, and enhancing model safety by creating preference data that steers the model away from harmful outputs (Chaffin et al., 2021; Liu et al., 2023; Khanov et al., 2024; Yu et al., 2024; Wang et al., 2023a; Singla et al., 2024; Li et al., 2024a; Zhang et al., 2025g; Yin et al., 2025). The methodology has also proven invaluable in scientific and highly specialized fields; for instance, it has been applied to accelerate discovery in catalyst design, improve diagnostic accuracy in medicine, create more strategic and proactive conversational agents, and master complex game environments (Guo et al., 2024; Volkova et al., 2024; Locowic et al., 2024; Sprueill et al., 2023; Light et al., 2024; Cheng et al., 2025; Tang et al., 2025; Ye et al., 2024; Ma et al., 2025; Park et al., 2024; Li & Ng, 2024; Du et al., 2024; Zheng et al., 2025; Duan & Wang, 2025; Jiang et al., 2025b; Pan et al., 2025b; Liu et al., 2025a; Zou et al., 2025; Garikaparthi et al., 2025; Shi et al., 2025c; Lu et al., 2025a). More recently, this data generation loop has been adapted for multimodal applications, generating high-quality visual reasoning trajectories to fine-tune Vision-Language Models (VLMs) and enhance their ability to solve complex multimodal problems (Wang et al., 2025b; Liu et al., 2025b; Du et al., 2025).

### 3.7 APPLICABILITY, TRADE-OFFS AND TASK-ORIENTED PRACTITIONER'S GUIDE

Our cross-paper analysis reveals consistent empirical patterns that determine when MCTS is most beneficial, how compute should be allocated, and how it compares to heuristic alternatives.

**Problem Suitability.** MCTS is most effective when *terminal rewards are reliable and deterministic*, enabling search to exploit combinatorial diversity without being overwhelmed by reward noise. This holds in mathematics and program synthesis, where unit tests or numeric checkers provide stable supervision (Qi et al., 2024; Zhang et al., 2024d). Recent work consistently reports large accuracy gains in such domains: ReST-MCTS* (Zhang et al., 2024b), rStar-Math (Guan et al., 2025), LLaMA-Berry (Zhang et al., 2025b), SVPO (Chen et al., 2024c), and LE-MCTS (Park et al., 2025) all show 10–40% improvements over greedy methods. By contrast, open-ended generation tasks lack verifiable correctness, and our synthesis shows that MCTS rarely exceeds $< 3\%$ improvement over beam search in such settings.

**Compute Allocation.** Two knobs dominate the compute–accuracy trade-off. (1) *Backup strategy*: *max* backups align with binary-verifier tasks (e.g., code) where discovering a single valid trajectory suffices, while *average* backups stabilize high-variance domains such as mathematics (Zhang et al.,

Table 2: Practitioner's Guide: Task-oriented MCTS configurations summarizing common structural choices and typical hyperparameter ranges.

| Task Domain | Topology | Evaluation | Backup | Typ. Hyperparams | Ref. Methods |
|---|---|---|---|---|---|
| **Math & Logic** | **Trace-based** *(Step or solution-level trees)* | **PRM / PPRM** or Self-Refine | Avg / Sum (value-driven) | $c_{puct} \in [1, 4]$ Rollouts: 16–128 Depth: 8–20 | ReST-MCTS* (Zhang et al., 2024b) rStar-Math (Guan et al., 2025) LLaMA-Berry (Zhang et al., 2025b) |
| **Code Generation** | **Terminal-state** *(Block/function-level)* | **ORM (execution)** + verbal feedback | Max (binary success) | Rollouts: 16–64 $k$ samples: 5–50 Temp: 0.6–0.8 | PG-TD (Zhang et al., 2023a) RethinkMCTS (Li et al., 2025c) |
| **RAG / Knowledge** | **Hierarchical** *(Retrieve → Reason)* | **Hybrid** (PRM + ORM) | Min / AND (weakest link) | Retrieval $k$: 3–10 Depth: 3–5 $\leq 10$ MCTS iters | RAG-Star (Jiang et al., 2025a) |
| **Autonomous Agents** | **State–Action** *(World-model tree)* | **Composite** (success + shaping) | Max-of-Avg (planning) | Depth: task horizon (typically 4–10) Rollouts: 20–50 High $c_{puct}$ | RAP (Hao et al., 2023a) LATS (Zhou et al., 2023) |

2024b; Li et al., 2025c). (2) *Evaluator cost*: high-fidelity PRMs/RMs (e.g., ReST-MCTS*, SVPO) reduce reward variance but shrink search depth; lighter-weight self-evaluation (e.g., MCTSr) supports deeper exploration. Across surveyed papers, allocating roughly 20–30% of the total budget to evaluation tends to yield robust improvements.

**PRM vs. ORM: When to Use Which?** PRMs provide fine-grained, step-level guidance and work well in **step-driven** or self-improvement frameworks such as ReST-MCTS* (Zhang et al., 2024b), SVPO (Chen et al., 2024c), rStar-Math (Guan et al., 2025), and LE-MCTS (Park et al., 2025). However, they require expensive annotation and may generalize poorly outside their domain. ORMs, in contrast, suit **goal-driven** tasks with verifiable terminal outcomes (e.g., code or mathematical checking), as used in RethinkMCTS (Li et al., 2025c) and RAG-Star (Jiang et al., 2025a). Their sparsity can cause shallow exploration or "false positives." Hybrid multi-critic designs (e.g., AlphaLLM (Tian et al., 2024a)) combine PRM shaping with ORM correctness, trading simplicity for stability.

**MCTS vs. Heuristic Search.** Heuristic search methods such as Tree-of-Thoughts (Yao et al., 2023) rely on LLM-generated intermediate heuristics and perform well when reasoning steps are interpretable and heuristics calibrated. MCTS instead accumulates *experience-driven* statistics (Hao et al., 2023a), making it better suited to sparse-reward or deceptive-intermediate regimes, common in long-horizon math proofs, code repair, and multi-step retrieval. Our review indicates that heuristic search excels under tight latency or when intermediate evaluation is reliable, whereas MCTS dominates when local plausibility diverges from global correctness.

Overall, MCTS is preferable when verifiable rewards exist or long-horizon dependencies matter; heuristic search fits tasks with strong intermediate heuristics or strict latency; and hybrid designs provide the best of both worlds in compositional tasks such as RAG or agent-based reasoning. Table 2 and Appendix E.2 summarizes recommended configurations across domains.

## 4 INFORMED SEARCH WITH LLM-GENERATED HEURISTICS

Informed search algorithms guide Large Language Model (LLM) reasoning by using heuristics to navigate vast problem spaces. Unlike classical methods with manually designed heuristics, modern approaches dynamically generate guidance using the LLM itself or auxiliary data. These methods primarily fall into two paradigms based on their heuristic design: direct state evaluation and composite A* cost functions. More details could be found in Appendix F.

This approach, exemplified by the Tree-of-Thoughts (ToT) framework (Yao et al., 2024b), uses an LLM as a direct, on-the-fly heuristic evaluator. First, an LLM generates multiple candidate next steps ("thoughts"). Then, a separate LLM-based evaluation assigns a heuristic value to each candidate. This score subsequently directs classical search algorithms, such as implementing a beam search (an informed BFS) to retain the top-$b$ states, or a pruned DFS to eliminate branches that fall below a certain value threshold.

A more sophisticated approach utilizes the A* search algorithm, which seeks to optimize the total cost function $f(n) = g(n) + h(n)$. This function balances the cost of the path taken so far, $g(n)$, with an estimated cost to reach the goal, $h(n)$. The primary innovation in methods like ToolChain*

(Zhuang et al., 2024) and Q* (Wang et al., 2024c) lies in constructing composite heuristics for $g(n)$ and $h(n)$ from diverse, LLM-relevant signals. The key components used to formulate these cost functions are summarized in Table 6.

Table 3: Compact Overview of A* Heuristic Components for LLMs

| Heuristic | A* Component | Mechanism (and Signal Source) |
|---|---|---|
| Process-Based Rewards | $g(n)$ | Aggregates step-wise rewards from execution feedback (e.g., logits, rule checks). |
| Statistical Consistency | $g(n)$ | Favors steps that are frequently proposed across multiple generation samples. |
| Memory-Based Comparison | $g(n), h(n)$ | Scores path similarity against a repository of high-quality examples (e.g., using LCS). |
| Learned Future Value | $h(n)$ | Estimates the cost-to-goal using a trained proxy model (e.g., a Q-function). |

## 5 EVALUATION FRAMEWORK AND COMPUTE PROTOCOLS

Recent advances in tree-structured decoding—e.g., MCTS-based reasoning (Xie et al., 2024a; Ha et al., 2025) and rStar-style agents (Guan et al., 2025)—show that test-time compute forms a scalable axis, often yielding a *model–search equivalence* where smaller models with search rival larger baselines. However, cross-paper comparisons remain infeasible due to heterogeneous assumptions about model size, evaluator cost, and hardware accounting (summarized in Appendix G).

To address this fragmentation, we propose the Standardized Compute-Reporting Protocol, a domain-agnostic framework designed to ensure comparability in future Tree-Search TTS research. A comprehensive definition of the protocol is provided in Appendix G.2. The core of SCRP involves decomposing the computational cost into a unified resource vector $\mathbf{B} = (C_{\text{policy}}, C_{\text{eval}}, C_{\text{verify}}, T_{\text{wall}})$. To facilitate hardware-agnostic comparison, we standardize the estimation of inference cost $\mathcal{C}_{\text{total}}$ for a problem instance $x$ as:

$$\mathcal{C}_{\text{total}}(x) \approx 2 \cdot P_{\text{policy}} \cdot T_{\text{policy}}(x) + 2 \cdot P_{\text{eval}} \cdot T_{\text{eval}}(x) + C_{\text{verify}}(x), \tag{3}$$

where $P$ denotes parameter counts and $T$ denotes token counts. Building on this abstraction, we recommend reporting **Budgeted Accuracy (Pass@FLOPs)** and **Tokens-per-Solved (TpS)** rather than raw accuracy alone, explicitly quantifying the trade-off between search depth, branching factors, and verification overhead.

## 6 CHALLENGES, FUTURE AND CONCLUSION

Despite clear gains in reasoning, tree-search methods face two major bottlenecks: **compute** and **reward quality**. Search introduces substantial overhead relative to greedy decoding (Wang et al., 2024a), exacerbated by strong models that often *overthink* simple queries (Chen et al., 2024d; Zeng et al., 2024a); structural constraints further limit parallelism and slow the self-play cycles required to distill search behavior into the base model (Xiang et al., 2025). Addressing these issues will require more adaptive and selectively activated search procedures with dynamic resource allocation and more aggressive pruning.

A second fundamental challenge is the difficulty of constructing reliable reward models. PRMs provide finer-grained supervision than ORMs but depend on costly, hard-to-scale annotations (Uesato et al., 2022; Lightman et al., 2023), and existing automated methods remain limited to narrow domains such as mathematics (Wang et al., 2024e; Luo et al., 2024). Imperfect rewards can misguide the search process and even induce *inverse inference scaling*, where additional rollouts degrade accuracy (Gao et al., 2023; Zeng et al., 2024b). The persistent gap between learned PRMs and oracle verifiers (Anonymous, 2024; Xiang et al., 2025) underscores the need for scalable methods to generate high-fidelity process rewards.

Our survey unified classical and MCTS-style methods around node representation, reward design, and algorithmic adjustments for LLMs. Future progress will depend on developing lighter-weight search dynamics and scalable, high-quality reward signals to fully realize tree search as a general-purpose reasoning mechanism.

# 7 REPRODUCIBILITY STATEMENT

This paper provides a survey and a unified perspective on tree search algorithms for Large Language Models (LLMs). As a review article, our primary contribution is the systematization and conceptual analysis of existing work, rather than the presentation of novel experimental results. To ensure the reproducibility of our analysis, we have based our survey exclusively on publicly available research papers. Every algorithm, framework, and concept discussed is explicitly cited, with full references provided in the bibliography. Our proposed taxonomy and unified notation, detailed in Section 3 and summarized in Tables 2 and 5, are derived directly from the methodologies described in these source publications. Readers can verify our classifications and synthesis by consulting the original papers, which form the basis for our claims. We have made every effort to accurately represent the works surveyed to ensure that our conceptual framework can be independently reviewed and validated by the research community.

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

APPENDIX

## A ORGANIZATION OF THE APPENDIX

The appendix is organized to provide a coherent, hierarchical progression from foundational concepts to methodological taxonomies, and finally to implementation-level guidance and evaluation standards. This structure is intended to support both conceptual understanding and practical adoption, enabling readers to navigate the diverse landscape of inference-time tree search through a unified lens. To address the complexity of the field, we utilize visual taxonomies and comparative tables to enhance skimmability. The supplementary material is divided into six modules:

**Foundational Paradigms (Appendix B):** We begin by revisiting three foundational search paradigms—uninformed search (BFS, DFS), informed search (heuristic-guided), and Monte Carlo Tree Search (learning from experience). These establish the algorithmic primitives, representational assumptions, and computational tradeoffs that underpin the subsequent design space and provide a common vocabulary for modern LLM-based adaptations.

**Theoretical Distinctions (Appendices C and D):**

- **Appendix C (Test-Time Optimization):** This section formalizes the shift from parameter-centric training to computation-centric inference. We introduce the notion of a *task-defined objective space*, decomposed into a *Prompt Space* (algorithm selection) and an *Answer Space* (solution generation). This framework provides the theoretical grounding for understanding MCTS as a form of structured test-time optimization.

- **Appendix D (Reward as Guidance vs. Learning Signal):** Here we provide a principled disentanglement of the "reward" construct. We contrast the persistent, parameter-updating role of reward in Reinforcement Learning with the transient, instance-specific role of reward in deliberative search. This distinction clarifies how inference-time reward shaping can guide reasoning without inducing long-term policy drift.

**Methodological Taxonomy (Appendices E and F):** These modules map the algorithmic design space underlying both MCTS and informed search.

- **Appendix E (Monte Carlo Tree Search):** We provide a comprehensive treatment of MCTS for LLMs, structured hierarchically to facilitate comparison. We first visualize the field's full typology and establish a unified notation. To aid practical adoption, we offer a **practitioner's guide** that synthesizes optimal search configurations across domains into a comparative summary. We then survey advanced topics and logically categorize applications into two distinct functional paradigms: *direct test-time enhancement* and *self-improvement* via synthetic data generation.

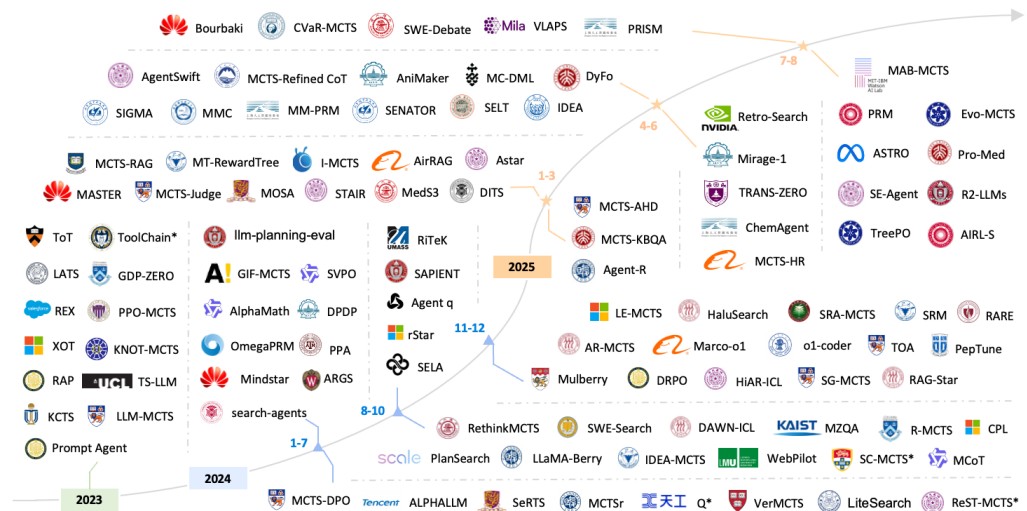

Figure 4: A map of the field's **rapid growth** on tree search algorithms.

- **Appendix F (Heuristic-Guided Search):** We analyze informed search methods—including LLM-augmented BFS/DFS and $A^*$—emphasizing heuristic construction, cost shaping, and admissibility tradeoffs. These techniques surface as complementary tools to MCTS within the broader space of inference-time reasoning.

**Standardized Evaluation Protocols (Appendix G):** To support reproducible and hardware-agnostic comparison, we propose a unified protocol for evaluating test-time compute. This includes practical recipes for FLOP estimation, wall-clock profiling, and rigorous metrics such as Budgeted Accuracy and Tokens-per-Solved, establishing a principled foundation for benchmarking search-based methods.

**Challenges and Future Directions (Appendix H):** We conclude with a discussion of emerging challenges, including overthinking behaviors on simple tasks, efficiency bottlenecks in deep search, and the heavy reliance on high-quality reward models. These issues motivate several directions for future research at the intersection of search, learning, and scalable inference.

The reorganized structure ensures that readers can first understand the conceptual axes and methodological design space before encountering algorithm-level details, eliminating the need to navigate long sequential listings.

## B  FOUNDATIONAL SEARCH PARADIGMS IN GENERAL AI

Solving complex problems can be formalized as a search task: finding an optimal path from an initial state to a goal state within a state-action space, conventionally represented as a tree $T_Q$. While classical AI has developed a rich toolkit for navigating such trees, the state spaces implicit in language model reasoning present unique challenges. They are not merely large; they are combinatorially vast, high-dimensional, and semantically structured, rendering exhaustive exploration computationally infeasible. This section revisits three foundational paradigms of tree search—uninformed, informed, and Monte Carlo-based—to establish a conceptual vocabulary for understanding their modern adaptations for LLM-based reasoning, where the goal is to identify optimal reasoning paths efficiently.

### B.1  UNINFORMED SEARCH: BLIND EXPLORATION

Traditional search algorithms, such as Breadth-First Search (Moore (1959), BFS), Depth-First Search (DFS), and Uniform Cost Search (UCS, or Dijkstra's algorithm), are **uninformed search** algorithms that operate with minimal knowledge about the goal. These algorithms can recognize the goal state when reached but lack any additional information to guide them toward it efficiently (Russell & Norvig, 2020; Poole & Mackworth, 2023). While some uninformed search algorithms, like UCS, consider the cost of the path taken so far, none can estimate the remaining distance to the goal or determine which paths are more promising.

The key characteristic of uninformed search is that it must rely solely on the problem's basic definition - the available actions, their costs, and the goal recognition criteria - to systematically explore the search space. As a result, these algorithms differentiate between possible solution paths primarily through their order of exploration and accumulated costs. Each algorithm offers different guarantees: BFS finds the shortest path in terms of steps, while UCS finds the lowest-cost path. Additional variants like Depth-Limited Search (DLS) and Iterative Deepening Search (IDS) address memory limitations of basic DFS while maintaining completeness. The choice between these algorithms often depends on the problem's characteristics and computational constraints, particularly memory requirements.

### B.2  INFORMED SEARCH: HEURISTIC-GUIDED EXPLORATION

**Informed search**, or **heuristic search**, refers to algorithms that leverage additional knowledge about the goal's location through domain-specific hints (Russell & Norvig, 2020). These hints are encoded in a **heuristic** function, denoted $h(n)$ (Poole & Mackworth, 2023):

$$h(n) = \text{estimated non-negative cost of the cheapest path from node } n \text{ to a goal state} \tag{4}$$

Table 4: A comparative taxonomy of foundational search algorithms in AI. Notation: $g(n)$ is the accumulated path cost to node $n$; $h(n)$ is the heuristic estimate of the cost from $n$ to the goal; $q_i$ is the estimated quality value of a search tree node $C_i$.

| Family | Algorithm | Guiding Signal / Principle | Typical Use Case |
|---|---|---|---|
| Uninformed | BFS | Explores layer-by-layer; guarantees shortest path in steps. | Shortest path, unweighted graphs. |
| | DFS | Explores a single branch to its depth before backtracking. | Path existence, memory efficiency. |
| | UCS | Expands node with the lowest accumulated path cost $g(n)$. | Optimal path, weighted graphs. |
| | IDS | Depth-first search with an incrementally increasing depth limit. | Optimal path, low memory overhead. |
| Informed | Greedy BeFS | Expands node closest to goal via heuristic $h(n)$ alone. | Quick, non-optimal solutions. |
| | A* Search | Balances path cost $g(n)$ & heuristic $h(n)$. | General-purpose optimal planning. |
| | Weighted A* | Biases toward heuristic via $g(n) + w \cdot h(n), w > 1$. | Speed-optimality trade-offs. |
| | IDA* | Iterative deepening applied to the A* cost function $f(n)$. | Memory-efficient optimal search. |
| | Beam Search | Keeps top-$k$ most promising candidates at each step. | High branching factor problems. |
| Monte Carlo (Sampling) | UCT-MCTS | UCT balances exploitation ($q_j$) & exploration. | Games/planning in vast state spaces. |
| | LLM-MCTS | LLM acts as policy prior $\pi$ and/or rollout policy. | Test-time deliberative reasoning. |
| | PUCT Variants | Integrates a policy network's prior $\pi$ into UCT bonus. | Integrating learned priors into search. |

Let $c(n, n')$ denote the cost of the path between nodes $n$ and $n'$. By incorporating heuristics, informed search algorithms can make educated decisions about which paths are most promising to explore, potentially reducing the computational resources required to find a solution. The effectiveness and properties of these algorithms depend critically on the quality of their heuristic functions. A heuristic is considered *admissible* if it never overestimates the true cost to the goal, and *consistent* if it satisfies the triangle inequality $h(n) \leq c(n, n') + h(n')$ for any successor $n'$ of $n$. The choice of heuristic function significantly impacts performance. A heuristic $h_1$ is considered more *informed* than $h_2$ if $h_1(n) \geq h_2(n)$ for all nodes $n$ and $h_1(n) > h_2(n)$ for some nodes. More informed heuristics generally lead to more efficient search, as they provide better guidance toward the goal.

However, there is often a trade-off between the computational cost of calculating the heuristic and the savings it provides in search efficiency. Common informed search algorithms include Greedy Best-First Search (BeFS), A* Search, Weighted A* Search, Iterative Deepening A* (IDA*), Beam Search, and Recursive Best-First Search (RBFS) . These algorithms vary in how they balance the heuristic estimates with path costs, leading to different trade-offs between optimality and efficiency. For instance, A* search, when used with an admissible heuristic, guarantees finding an optimal solution if one exists. The success of these algorithms in practical applications often depends on designing effective problem-specific heuristics. Common techniques for developing heuristics include relaxing problem constraints, pattern databases, and learning from experience (Russell & Norvig, 2020). While informed search algorithms generally outperform uninformed search in practice, their effectiveness relies heavily on the quality of their heuristic functions and the specific characteristics of the problem domain.

### B.3    MONTE CARLO TREE SEARCH: LEARNING FROM EXPERIENCE

Monte Carlo Tree Search (MCTS) was first introduced by Coulom (2006) in the context of computer Go as an **adversarial search** algorithm, which aims to maximize winning probability against

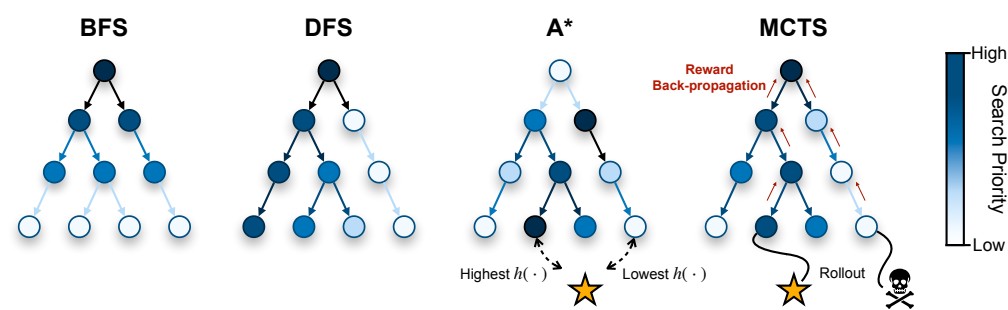

Figure 5: A visual comparison of four fundamental tree search algorithms, where node color intensity represents search priority. **BFS** explores exhaustively level by level, while **DFS** commits to a single path until a leaf is reached. In contrast, informed search like **A\*** uses a heuristic function $h(\cdot)$ to prioritize nodes with the lowest estimated total cost, regardless of their depth. **MCTS** introduces a statistical approach, using simulated rollouts from leaf nodes and backpropagating the outcomes to dynamically guide the search toward high-reward regions of the tree.

an optimal opponent. While adversarial MCTS alternates between players and models opponent responses, the MCTS variant used in LLM's inference-time search is a *single-agent* formulation, where the algorithm explores different action sequences without modeling opposing players. This adaptation maintains MCTS's core strengths in balancing exploration and exploitation through statistical sampling, while refocusing the objective from competitive game-playing to finding optimal sequences of actions in a non-adversarial environment.

Inference-time MCTS (hereafter referred to simply as MCTS) retains the four fundamental phases of the original algorithm: selection, expansion, simulation, and backpropagation. During selection, the algorithm traverses the tree using the *Upper Confidence bounds applied to Trees (UCT) policy*, which balances exploration and exploitation by selecting nodes (states) that maximize:

$$a^* = \arg \max_{a \in A(s)} \left[ Q_i + c\sqrt{\frac{\ln n_i}{N(s,a)}} \right] \tag{5}$$

where $Q(s,a)$ estimates the expected future reward of taking action $a$ in node $s$, $N(s)$ is the number of times node $s$ has been visited, $N(s,a)$ is the number of times action $a$ has been selected in node $s$, $c$ is an exploration constant, and $A(s)$ is the set of available actions at node $s$ (Kocsis & Szepesvari, 2006). In the expansion phase, new nodes sampled by LLMs (e.g. subsequent steps in reasoning) are added to the tree to gradually build a model of the search space. The simulation phase performs rollouts from leaf nodes using a default policy to estimate long-term rewards, replacing the win/loss outcomes of adversarial MCTS with domain-specific reward measures.

Unlike traditional uninformed search algorithms such as BFS or DFS that systematically explore the state space, MCTS offers a statistical sampling approach that can handle much larger search spaces. Compared to informed search algorithms like A\*, which rely on pre-defined heuristics, MCTS builds its evaluation function through experience. This makes it particularly suitable for LLM inference where defining accurate heuristics is challenging. The algorithm's ability to balance between exploration and exploitation, combined with its flexibility in handling large state spaces, makes it a powerful tool for guiding LLM inference, though its effectiveness depends on carefully managing the trade-offs between computational resources and search depth.

### B.4 COMPARISON OF EXPLORATION STRATEGIES

Figure 5 provides a conceptual illustration of these distinct exploration strategies. Uninformed algorithms like BFS and DFS are governed by rigid, topology-driven expansion protocols. Informed search, exemplified by A\*, introduces goal-directedness by prioritizing search based on a heuristic cost-to-go estimate, $h(\cdot)$, allowing it to focus on promising regions irrespective of tree topology. Finally, MCTS replaces the static heuristic with a dynamically learned value function, estimated via

statistical sampling. This adaptive, self-correcting mechanism allows it to focus computational resources on the most promising regions of the search space without requiring prior domain knowledge encoded in a heuristic. This very property makes it the preeminent search paradigm for navigating the vast and ill-defined reasoning spaces of large language models.

# C  TEST-TIME SCALING VIA SEARCH

As the scaling of model parameters and training data yields diminishing returns, a new frontier has emerged: **test-time scaling**. This paradigm investigates how to optimally allocate computational resources during inference to enhance a model's effective reasoning capabilities. Unlike training-time scaling, which refines a global, amortized policy by encoding knowledge into a model's weights, test-time scaling performs instance-specific optimization for a given problem $Q$. This section provides a detailed, mathematically-grounded analysis of these two orthogonal paradigms, contrasting how they operate in fundamentally different optimization landscapes: the latent parameter space for training versus the task-defined objective space for inference.

## C.1  A TALE OF TWO OPTIMIZATIONS FOR LLM SCALING: TRAINING-TIME VS. TEST-TIME

The figure referenced illustrates two distinct approaches for improving model performance, each defined by its unique objective signal and the space over which it optimizes.

**Training-Time Scaling: Optimization in Latent Parameter Space.**  During training, the primary goal is to learn a set of parameters $\theta^*$ that minimizes an expected loss function $\mathcal{L}$ over a data distribution $\mathcal{D}$. The optimization problem is formally stated as:

$$\theta^* = \arg\min_{\theta \in \Theta} \mathbb{E}_{(i,o) \sim \mathcal{D}}[\mathcal{L}(f_\theta(i), o)],$$

where $\Theta \subseteq \mathbb{R}^N$ is the high-dimensional **latent parameter space**. The **objective signal** in this paradigm is the gradient of the loss with respect to the parameters, $\nabla_\theta \mathcal{L}$. Optimization proceeds via iterative updates, such as stochastic gradient descent. The result is a static artifact—a trained model $\pi$—that implicitly represents a posterior distribution over solutions.

**Test-Time Scaling: Optimization in Task-Defined Objective Space.**  Given a fixed, pretrained model $\pi$, test-time scaling seeks to find an optimal reasoning trace $p^*$ for a specific problem instance $Q$. This process constitutes a second, distinct optimization loop. The search occurs in a discrete, structured **task-defined objective space**, the solution space $\mathcal{P}(Q)$, which consists of all possible reasoning traces. The **objective signal** is a scalar **reward** or **value** that evaluates the quality of a trace. The optimization problem at inference is therefore:

$$p^* = \arg\max_{p \in \mathcal{A}(\pi, Q, \mathcal{C}_{\text{infer}})} V(p),$$

where $\mathcal{A}(\pi, Q, \mathcal{C}_{\text{infer}})$ is the search algorithm that explores a subset of $\mathcal{P}(Q)$ guided by the model's prior $\pi$ and constrained by the inference compute budget $\mathcal{C}_{\text{infer}}$, and $V(p)$ is a function evaluating the final trace. Scalable inference techniques, such as tree search, use intermediate rewards $r_s$ or partial trace values $v_i$ to dynamically allocate compute to more promising regions.

## C.2  OPERATIONALIZING SEARCH IN THE OBJECTIVE SPACE

The conceptual shift from gradients in latent space to rewards in objective space necessitates a different class of optimization algorithms. While training relies on gradient-based methods, test-time scaling is operationalized by search procedures that can navigate complex, non-differentiable solution spaces.

**Tree Search as a Scalable Inference Optimizer.**  Tree search methods, particularly MCTS, provide a principled framework for this optimization. They build a search tree $T_Q$ where each node $C_i$ corresponds to a partial reasoning trace $p_i$. At each node, an action selection policy balances exploiting known high-reward paths and exploring novel ones. For LLM-based search, this policy

often uses a PUCT-style rule that incorporates the policy network's prior. The next action $a^*$ is selected by choosing the action that leads to the most promising child node:

$$a^* = \arg\max_{a \in \mathcal{A}(s_i)} \left( q_j + U(C_i, C_j) \right),$$

where $s_i$ is the state at the parent node $C_i$, and action $a$ leads to the child node $C_j$ with quality value $q_j$. The uncertainty bonus $U(C_i, C_j)$ is formulated as:

$$U(C_i, C_j) = c_{\exp} \cdot \pi(a|p_i, Q) \cdot \frac{\sqrt{n_i}}{1 + n_j}.$$

Here, $n_i$ and $n_j$ are the visit counts of the parent and child nodes, respectively. The policy $\pi$ provides a prior probability for taking action $a$ given the history $p_i$, and $c_{\exp}$ is an exploration hyperparameter. This synthesis allows the algorithm to scale reasoning performance effectively with the allocated inference compute budget.

### C.3 DECOMPOSING THE OBJECTIVE SPACE: PROMPT AND ANSWER SPACES

The task-defined objective space, over which test-time search operates, is not monolithic. It can be productively decomposed into two distinct, hierarchically-related search spaces: the **Prompt Space** and the **Answer Space**. This decomposition clarifies the mechanisms of Chain-of-Thought (CoT) reasoning and reveals the limitations of many current test-time search methods. The overall optimization problem is thus a search for an optimal reasoning trace, which involves finding both the right algorithm and its correct execution.

**The Prompt Space ($\mathcal{P}$): Searching for an Algorithm.** The prompt space, $\mathcal{P}$, encompasses the set of all possible reasoning structures or "step templates" an LLM can adopt to solve a problem. Each template $p \in \mathcal{P}$ represents a specific strategy for externalizing and manipulating information from the model's latent state $\mathbf{h}$ into its textual output space (Zhang et al., 2025e). In essence, selecting a template $p$ is equivalent to selecting an **algorithm**. For example, one template for a complex arithmetic task might involve explicitly tracking a running total, while another might only verbalize intermediate calculations without a canonical state representation.

The choice of template is paramount because it dictates the computational graph the model simulates through its autoregressive generation. While theoretical work suggests that a CoT-augmented Transformer can be Turing-complete (Li et al., 2024d), this potential is contingent on generating the correct computational trace. An suboptimal template can lead to an inefficient or even intractable search by failing to surface the necessary state information for subsequent steps, effectively breaking the simulated recurrence. The search for an optimal $p^* \in \mathcal{P}$ is therefore a meta-level optimization: discovering the most effective procedure for solving the task instance.

**The Answer Space ($\mathcal{S}$): Searching for a Solution.** For any given prompt template $p$, there exists a corresponding answer space, $\mathcal{S}_p$, which contains all possible reasoning traces (i.e., potential solutions) that can be generated by adhering to that template's structure. The complexity of navigating this space is critically conditioned on the choice of $p$. An effective template $p^*$ dramatically prunes the answer space, simplifying the path to a correct solution. Conversely, a poorly chosen template $p'$ can render the answer space vast and unstructured, making the search computationally infeasible even with a large compute budget.

Many contemporary test-time search methods, such as Tree-of-Thought (Yao et al., 2024c) and Graph-of-Thought (Besta et al., 2024), operate primarily within this second level of the hierarchy. They typically fix a single, heuristically-defined prompt template (e.g., via a generic instruction like "think step by step") and then deploy sophisticated search algorithms to navigate the resulting answer space $\mathcal{S}_p$. These approaches excel at mitigating execution errors and exploring diverse solution paths *within a fixed algorithmic strategy*. However, they do not address the foundational challenge of selecting the algorithm itself. If the governing template $p$ is flawed, even an exhaustive search of $\mathcal{S}_p$ is unlikely to yield a correct solution.

**A Unified View of Test-Time Search.** A comprehensive framework for test-time search must therefore account for the joint optimization over both spaces. The ultimate objective is to discover a

solution trace $s^*$ that maximizes the value function $V(\cdot)$, where the search spans all possible traces allowed by all possible templates:

$$s^* = \arg \max_{p \in \mathcal{P}, s \in \mathcal{S}_p} V(s)$$

This formulation highlights a critical gap in current research. While significant effort has been invested in optimizing search algorithms within a given answer space $\mathcal{S}_p$, the systematic exploration of the prompt space $\mathcal{P}$ remains a largely open challenge. The true potential of test-time scaling lies not merely in executing a known algorithm more robustly, but in dynamically discovering the most effective algorithm for the specific problem at hand.

## D  REWARD AS A UNIFIED SIGNAL FOR RL AND SEARCH : ONE OBJECTIVE, TWO OPTIMIZERS

In advanced AI systems, a reward signal is the fundamental currency for guiding behavior. However, its role bifurcates into two distinct yet complementary functions depending on the temporal scope of the objective: shaping a durable, long-term **policy** versus guiding a transient, short-term **plan**. This distinction is not one of paradigm but of application—whether the reward is used to permanently update the model's internal parameters (RL learning) or to direct a temporary search with fixed parameters (planning).

### D.1  RL VIA POLICY SHAPING: INTERNALIZING REWARDS FOR GENERALIZATION

When a reward signal is coupled with a learning algorithm, such as in Reinforcement Learning (RL), its purpose is to be **internalized**. The feedback from the reward directly modifies the model's weights, creating lasting changes in its behavior. This process is analogous to skill acquisition, where experience is distilled into a robust, general-purpose policy that governs the agent's "instincts" across all future tasks. Formally, this involves optimizing policy parameters $\theta$ to maximize an objective $\mathcal{J}_{\text{RL}}$ that integrates task rewards with adherence to a set of universal principles $\mathcal{P}$.

The optimization objective can be expressed as finding the optimal parameters $\theta^*$ that balance expected cumulative rewards $G(\tau)$ over trajectories $\tau$ with a regularization term that enforces alignment with a foundational policy prior $\pi_{\mathcal{P}}$:

$$\theta^* = \arg \max_{\theta} \mathbb{E}_{\tau \sim \pi_\theta} \left[ G(\tau) \right] - \lambda \int_{s \in \tau} D_{KL} \left( \pi_\theta(\cdot|s) \| \pi_{\mathcal{P}}(\cdot|s) \right) ds$$

where $D_{KL}$ is the Kullback-Leibler divergence, measuring the "cost" of deviating from the ingrained principles, and $\lambda$ is a hyperparameter controlling the strength of this alignment imperative. Because this learning is permanent, the reward function is designed to instill **universal, foundational principles**—for example, promoting logical consistency, ensuring truthfulness, or encouraging methodical reasoning. The objective is not to solve a single problem but to forge a broadly capable and aligned agent. The reward here acts as a long-term teacher, shaping the agent's intrinsic character for future, unseen challenges.

### D.2  SEARCH VIA DELIBERATIVE PLANNING: EXTERNALIZING REWARDS FOR SPECIFICITY

Conversely, during test-time search, the reward signal functions as an **external, ephemeral guide**. It directs a deliberative process, like Monte Carlo Tree Search (MCTS), to navigate the solution space for a single, immediate task. The reward evaluates candidate action sequences (plans), allowing the system to identify a high-quality solution for the specific problem at hand. For a given task with a specific external reward function $R_{\text{ext}}$, the goal is to find an optimal plan $p^*$ that maximizes a combination of this external signal and an internal, path-dependent heuristic $\mathcal{H}_\theta$ provided by the frozen model.

The optimal plan $p^*$ for a state sequence $s_0, s_1, \ldots, s_T$ resulting from the plan's actions is found by solving:

$$p^* = \arg \max_{p \in \mathcal{P}_{\text{plan}}} \left[ \sum_{t=0}^{T-1} \gamma^t R_{\text{ext}}(s_t, a_t) + \mathcal{H}_\theta(s_T, p) \right]$$

where the heuristic $\mathcal{H}_\theta$ is not just a simple state evaluation but a complex function of the final state $s_T$ and the path $p$ taken, potentially incorporating penalties for path irregularity or deviation from the model's learned priors:

$$\mathcal{H}_\theta(s_T, p) = V_\theta(s_T) - \beta \cdot \log\left(\int_{\tilde{p}\in\mathcal{N}(p)} e^{-\mathcal{E}(\tilde{p})/\tau_c} d\tilde{p}\right)$$

Here, $V_\theta(s_T)$ is the model's intrinsic value estimate, while the second term acts as a complexity penalty based on the "free energy" over a neighborhood of paths $\mathcal{N}(p)$, discouraging overly surprising or convoluted solutions. Crucially, this feedback is discarded once the task is complete; the model's underlying parameters $\theta$ remain untouched. This makes the reward an ideal tool for **task-specific, localized objectives** without the risk of corrupting the model's general-purpose policy.

### D.3 A Symbiotic Framework

Ultimately, policy shaping and deliberative planning are not competing methodologies but two integrated components of a sophisticated decision-making architecture. The RL-trained policy provides the foundational intuition, offering high-quality, pre-compiled heuristics that make the search space tractable. Search then provides the focused deliberation needed to refine these intuitions into a precise plan for the current context. This symbiotic relationship can be captured in a single, bi-level optimization objective, where the outer loop learns the policy parameters $\theta$ by anticipating the outcome of the inner-loop search process over a distribution of tasks $\mathcal{I} \in \mathcal{D}$.

The overarching goal is to find policy parameters $\theta^*$ that maximize the true, ground-truth reward $R_{\text{true}}$ of the plans generated by the search algorithm:

$$\theta^* = \arg\max_\theta \mathbb{E}_{\mathcal{I}\sim\mathcal{D}}\left[R_{\text{true}}\left(\arg\max_{p\in\mathcal{P}_{\text{plan}}}\left\{\sum_{t=0}^{T-1}\gamma^t R_{\text{ext},\mathcal{I}}(s_t, a_t) + \mathcal{H}_\theta(s_T, p)\right\}\right)\right]$$

This formulation reveals the deep connection between the two processes. The outer optimization (learning) seeks to create a model whose internal heuristic, $\mathcal{H}_\theta$, is maximally useful for the inner optimization (planning), which in turn must produce plans that score well on the final, external metric $R_{\text{true}}$. In essence, one process builds the artist's foundational skill over a lifetime, while the other guides the brushstrokes for the single masterpiece they are creating now.

## E  Monte Carlo Tree Search (MCTS)

### E.1 Unified Notation and Problem Formation

We adopt the notation conventions introduced in ReST-MCTS* (Zhang et al., 2024a) to formalize MCTS in the context of LLM reasoning in a *unified manner*. This approach ensures that all the articles surveyed adhere to a *consistent notation system* (with minor adjustments to accommodate unique designs), allowing for a clear comparison of their methods without the reader having to navigate the discrepancies in notation.

We first introduce the table of notations:

With this set of notations defined, a search problem in LLM based reasoning can be generalized as finding the correct solution *or* the optimal reasoning trace $p' = [s'_1, s'_2, \cdots, s'_n]$ for a given problem $Q$.

We categorize approaches for finding correct final solution (a specific terminal state $s'$) as goal-driven. Goal-driven methods focus primarily on arriving at the correct final answer for given reasoning problems, paying less attention to the reasoning trace that leads to it. In contrast, approaches that aim to identify good or optimal reasoning steps for a given problem are categorized as step-driven. Step-driven methods not only seek to find the correct solution but also emphasize discovering high-quality intermediate steps that contribute meaningfully to the reasoning process and minimize the reasoning distance.

In the search process, the reasoning LLM acts as a policy network $\pi(\cdot|Q, c)$. where it generates a sequence of reasoning steps or actions to solve the problem $Q$, under a given instruction prompt $c$. The

| Symbol | Definition |
|---|---|
| $Q$ | Input question or problem for which reasoning is being performed |
| $c$ | User prompt or conditioning input used to bias the reasoning traces |
| $a_i$ | Reasoning action at step $i$ generated by the LLM (policy network), where $a_i \in \mathcal{A}$ |
| $s_i$ | Reasoning state at step $i$ resulting from action $a_i$ |
| $p_i$ | Partial reasoning trace up to step $i$, defined as $p_i = [s_1, s_2, \ldots, s_i]$ |
| $r_{s_i}$ | Single-step reward for state $s_i$, measuring its quality independent of previous states |
| $v_i$ | Value of partial solution $p_i$, indicating its potential to reach a correct final answer |
| $T_Q$ | Search tree for problem $Q$, where each node uniquely identifies a reasoning trace |
| $\pi$ | Policy model (LLM) used to generate reasoning steps during tree search |
| $V_\theta$ | Value model that computes partial trace values: $v_i = V_\theta(p_i)$ |
| $R_\theta$ | Reward model that generates single-step rewards: $r_{s_i} = R_\theta(s_i)$ |
| $\mathcal{A}$ | Action space available at state $s_i$, representing all possible next actions |
| $C_i$ | **Search tree node**, represented as $C_i = (t_i, n_i, q_i)$ where: 
 • $t_i$: tree node that identifies $C_i$ 
 • $n_i$: Visit count of node $C_i$, tracking exploration frequency 
 • $q_i$: Quality value of the partial solution at node $C_i$, indicating its potential to lead to a correct answer |

Table 5: Unified Notations for MCTS-Based Methods in LLM Reasoning

sequence of generated state-action pairs by $\pi(\cdot|Q, c)$ is denoted as $[s_1, a_1, s_2, a_2, s_3, a_3, \cdots, s_n]$, where $s_1$ is the initial state (often a dummy answer or system prompt) and $s_n$ is the terminal state. The terminal state $s_n$ is reached when [eos] (i.e. end of sequence) token is produced, which may signify the generation of a final answer (correct or incorrect) or the exhaustion of the step limit (e.g. max context length).

Note that, unlike most other reinforcement learning (RL) problems, where an action $a_i$ leads to different states $s_{i+1}$ based on a state transition probability, a reasoning action $a_i$ in LLM-based reasoning deterministically leads to a fixed next reasoning state. This deterministic nature is due to the structure of reasoning (with rare exceptions). As a result, we clarify the usage of certain notations, which may differ from those in typical RL formulations:

- A reasoning trace, or partial solution, $p_i$, can be expressed in two equivalent forms:

$$p_i = [s_1, a_1, s_2, a_2, s_3, a_3, \ldots, s_i]$$

or

$$p_i = [s_1, s_2, s_3, \ldots, s_i].$$

The first form treats actions as distinct from states, while the second combines actions and resulting states into $s$. There is no inherent difference between the two representations, as LLM outputs both $s_i$ and $a_i$ into a sentence in each reasoning step during Chain of Thought. Some looks at it separately (such as RAP) while others take a joint view (such as ReST-MCTS*).

- Unlike traditional RL, where the reward is calculated based on the state-action pair, denoted as $R(a, s)$, and depends on the different state transitions resulting from action $a$, the reward of a single LLM reasoning step can be evaluated based on either the action $a_i$ or the resulting state $s_{i+1}$, or even on state action pairs $(s, a)$, due to the deterministic nature of reasoning (each $a$ deterministically determines $s$).

For *simplicity*, we typically consider $s_i$ to be a natural language sentence generated as one chain-of-thought (CoT) reasoning step. Consequently, $p_i = [s_1, s_2, s_3, \ldots, s_i]$ represents a CoT trace consisting of $i$ sentences generated in $i$ sequential steps by LLMs.

During reasoning, a given reasoning state $s_i$ can transition to different next reasoning states $s_{i+1}$, deterministically, depending on the different action $a_i$ that is chosen (from the action space $\mathcal{A}$) by the LLM policy $\pi$, forming a tree structure, denoted as $T_Q$.

Monte Carlo Tree Search (MCTS) optimizes the search for the reasoning trace $[s_1, s_2, \ldots, s_n]$ in $T_Q$ to find correct answers. Each partial solution trace $p_i = [s_1, s_2, \ldots, s_i]$ forms a unique path (or even node) in this tree, associated with its estimated value $v_i$ and visit count $n_i$. The value $v_i$ defines how promising such partial trace is to reach the correct answer. MCTS process is guided by this promising indicator $v_i$.

Unsurprisingly, the design and computation of $v_i$ become one of the most critical challenges in search algorithm design for LLM reasoning. Our survey places particular emphasis on the methods used to design the value function $V(\cdot)$ in each of the surveyed papers.

All of the search to be discussed here is done in *Answer Space* of problem $Q$, for the discussion of searching in *Prompt Space* of LLM, refer to Section.

### E.2 PRACTITIONER'S GUIDE: TASK-ORIENTED MCTS GUIDE

We observe that optimal search configurations—specifically node granularity, evaluation signals, and backpropagation logic—are distinct functions of the task domain's reward sparsity and error propagation characteristics. Table 2 synthesizes these domain-specific primitives.

**Mathematical Reasoning: Mitigating Variance via Trace-Based Search.** In mathematical domains, the primary challenge is *error accumulation*, where a single logical fault invalidates the subsequent trajectory. Consequently, relying solely on Outcome Reward Models (ORMs) induces high variance due to "false positives" (correct answers derived from flawed reasoning).

- **Topology & Evaluation:** We recommend **Trace-based nodes** ($p_i = [s_1 ... s_i]$), enabling the value function to condition on the full derivation history rather than the immediate state. Evaluation should leverage **Process Reward Models (PRMs)** to verify intermediate steps. In the absence of trained PRMs, methods like **MCTSr** effectively substitute the model with LLM-based self-refinement.
- **Backup Dynamics:** The objective is *robustness*. Practitioners should employ **Average** or **Sum** backup rules rather than Maximization. A reasoning path is only reliable if the density of correct rollouts is high, thereby filtering out lucky guesses.

**Code Generation: Exploiting Binary Oracles.** Code generation is distinct from reasoning due to the availability of a deterministic oracle (the compiler/test suite). The search objective shifts from maximizing expected utility to ensuring the *existence* of a solution.

- **Topology & Evaluation: Terminal-State nodes** are sufficient, as the intermediate logic is often opaque until execution. The primary signal is **Execution Feedback (ORM)**. Advanced implementations (e.g., **RethinkMCTS**) integrate verbal feedback from failed tests into the prompt state for subsequent iterations.
- **Backup Dynamics:** Because the reward signal is binary (pass/fail), **Max** backup updates ($Q_i \leftarrow \max(Q_i, r_{new})$) are optimal. Finding a single passing solution satisfies the task requirements; the average quality of failed attempts is irrelevant to the final utility.

**RAG & Knowledge Tasks: The Weakest Link Principle.** Knowledge-intensive tasks require a strict logical conjunction between retrieval relevance and answer correctness. A high-fidelity answer derived from irrelevant documents constitutes a hallucination.

- **Topology & Evaluation:** The search space should be modeled via **State-Action nodes** explicitly separating "Retrieval" actions from "Reasoning" actions. Evaluation demands a **Hybrid** signal: a PRM for document relevance and an ORM for factual consistency.
- **Backup Dynamics:** To enforce factual integrity, we recommend **Min-based aggregation** ($V(s) = \min(r_{steps})$), as utilized in HiAR-ICL. This enforces a "weakest link" logic, ensuring that a hallucination or retrieval failure in any single step penalizes the value of the entire reasoning chain, preventing the propagation of grounded but irrelevant text.

**Autonomous Agents: Lookahead in Latent World Models.** Agents operate in partially observable environments where actions induce irreversible state transitions. MCTS here serves as a planner using the LLM as a simulator.

- **Topology & Evaluation:** Nodes must represent **State-Action pairs** $(s_t, a_t)$, where the LLM functions as a **World Model** predicting $s_{t+1}$. Effective rewards are composite: $r_t = r_{prob}^{\alpha} \cdot r_{utility}^{1-\alpha}$, balancing the prior likelihood of an action (naturalness) with its task utility.

- **Backup Dynamics:** Given the long search horizons, getting stuck in local optima is a significant risk. Practitioners should increase exploration constants ($c_{puct}$) and employ **Max of Averages** for backup, isolating the single best plan from a diverse set of simulations.

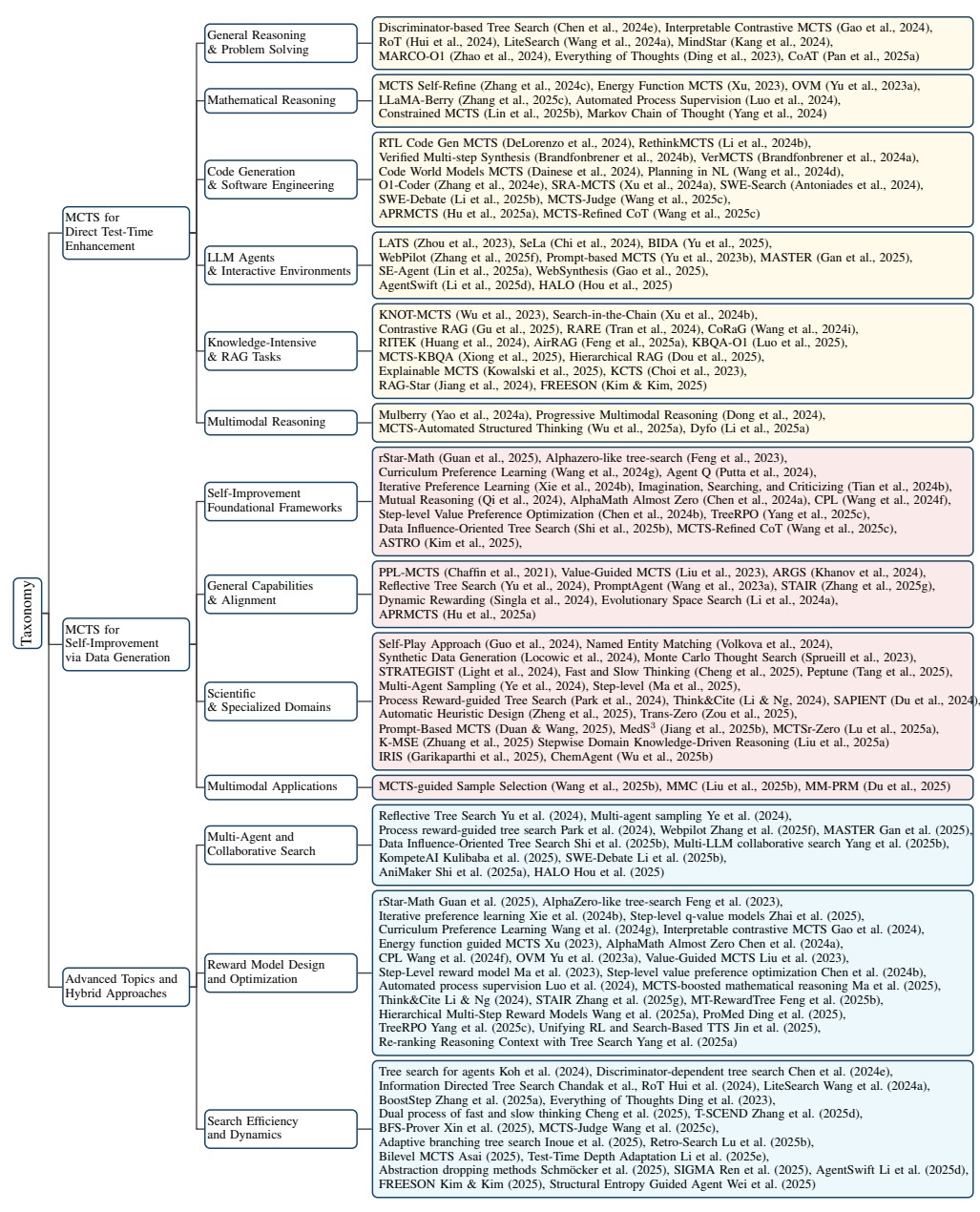

Figure 6: A comprehensive taxonomy of MCTS.

### E.3    ADVANCED TOPICS AND HYBRID APPROACHES FOR MCTS

As the field matures, researchers are exploring more sophisticated techniques that refine the core search paradigm, create better reward signals, and combine multiple methodologies.

#### E.3.1    MULTI-AGENT AND COLLABORATIVE SEARCH

Instead of a single LLM performing a search, these approaches use multiple LLM agents that collaborate, debate, or take on specialized roles to solve a problem more effectively. This paradigm shifts from a monolithic searcher to a coordinated team, enabling more robust and diverse problem-solving. For instance, some frameworks use MCTS to orchestrate multiple agents, dynamically adjusting their number and communication based on task complexity Gan et al. (2025). Others employ hierarchical structures with specialized agents for high-level planning, role design, and low-level execution Hou et al. (2025). In competitive settings, such as software issue resolution, multi-agent debate frameworks encourage diverse reasoning paths and lead to more consolidated solutions Li et al. (2025b). Another collaborative approach, the Mixture-of-Search-Agents (MOSA), leverages the collective expertise of multiple LLMs by combining their independent explorations with iterative refinement, which helps mitigate the limitations of any single model Yang et al. (2025b).

#### E.3.2    REWARD MODEL DESIGN AND OPTIMIZATION

The success of any search algorithm hinges on the quality of its reward function. This area focuses on designing, training, and analyzing reward models that can accurately guide the search process. A significant trend is the shift from coarse, outcome-based rewards to more granular, step-level feedback. Process-Supervised Reward Models (PRMs) provide this step-by-step guidance, improving reasoning in tasks like mathematics and code generation Ma et al. (2023). However, annotating these steps is costly, leading to automated data collection pipelines that use MCTS to generate large-scale, step-level supervision data efficiently Luo et al. (2024). Research also explores alternatives, such as Outcome-supervised Value Models (OVMs), which are trained only on final outcomes but effectively learn to assess the potential of incomplete reasoning paths, acting as a value function for planning Yu et al. (2023a). More advanced hybrid approaches unify reinforcement learning and search by demonstrating that a reward function learned via RL can serve as an ideal PRM for guiding search, eliminating the need for labeled process data Jin et al. (2025). Other innovations include Hierarchical Reward Models (HRMs) that evaluate both individual steps and their coherence in sequence Wang et al. (2025a) and comprehensive frameworks for building domain-specific reward models, such as for machine translation Feng et al. (2025b). Analysis of these models reveals counterintuitive findings; for instance, step-level reward models are more adept at assessing the logical coherence of mathematical language than the nuances of natural language descriptions Ma et al. (2025).

#### E.3.3    SEARCH EFFICIENCY AND DYNAMICS

A major challenge for tree search is its high computational cost. These works focus on making the search process more efficient and adaptive. To reduce wasted computation, methods like LiteSearch introduce dynamic node selection and node-level exploration budgets based on guidance from a value network Wang et al. (2024a). Algorithmic enhancements, such as bilevel MCTS, can achieve amortized $O(1)$ runtime for node selection, significantly speeding up planning in domains with deep search trees Asai (2025). Another strategy is to guide the search more intelligently; Information Directed Tree Search (IDTS), for example, uses a Bayesian approach to quantify the information gain from different feedback types, steering the search toward more informative paths Chandak et al.. The search process can also be made more dynamic and adaptive. Adaptive Branching MCTS (AB-MCTS) dynamically decides at each node whether to "go wider" by expanding new candidates or "go deeper" by refining existing ones, effectively generalizing repeated sampling Inoue et al. (2025). Some approaches even adapt the model's architecture at inference time, creating a custom "chain-of-layers" for each sample by skipping or repeating layers from the pretrained model as needed Li et al. (2025e). Other works focus on improving the quality of reasoning within the search; BoostStep, for instance, enhances single-step reasoning through a step-aligned in-context learning mechanism that provides more relevant examples Zhang et al. (2025a). For MCTS variants that use

abstractions to simplify the search space, new methods have been proposed to dynamically drop these abstractions in time-critical settings to ensure optimal performance Schmöcker et al. (2025).

### E.4 MCTS FOR DIRECT TEST-TIME ENHANCEMENT

This category includes methods that use Monte Carlo Tree Search (MCTS) primarily to improve the quality of the LLM's output for a single, given prompt at inference time, without updating the model's weights. These approaches treat the generation of a solution as a sequential decision-making problem, where the MCTS algorithm explores a tree of possible reasoning steps or text segments to find an optimal path. The core idea is to leverage lookahead planning to overcome the greedy, left-to-right nature of standard autoregressive decoding, thereby enhancing the model's performance on tasks that require strategic thinking, exploration, or backtracking.

#### E.4.1 GENERAL REASONING & PROBLEM SOLVING

This area focuses on creating domain-agnostic frameworks to enhance the fundamental reasoning capabilities of LLMs. Research here aims to make MCTS-based inference more efficient, interpretable, and robust. For instance, some works seek to improve search efficiency by designing more lightweight algorithms or dynamic resource allocation strategies, reducing the substantial computational overhead typically associated with tree search Wang et al. (2024a); Gao et al. (2024). Others incorporate meta-cognitive strategies like reflection, where the model learns from previous search experiences within the same problem to avoid repeating mistakes, effectively summarizing successful strategies to guide future steps Hui et al. (2024). Another line of inquiry investigates the core components and limitations of tree search, finding that its effectiveness is often contingent on the accuracy of a reward model or discriminator that evaluates intermediate steps Chen et al. (2024e). To broaden the search space and emulate human-like associative thinking, methods like Chain-of-Associated-Thoughts (CoAT) integrate MCTS with dynamic memory modules, allowing the model to incorporate new information during the reasoning process Pan et al. (2025a). These general-purpose enhancements treat complex problem-solving as a formal search task, building frameworks that integrate external knowledge and planning capabilities to handle open-ended challenges Ding et al. (2023); Zhao et al. (2024); Kang et al. (2024).

#### E.4.2 MATHEMATICAL REASONING

Mathematics provides an ideal testbed for MCTS because its problems have clear, verifiable solutions, which simplifies the design of effective reward functions. This verifiability allows for precise feedback on the correctness of intermediate reasoning steps or the final outcome. Many approaches in this domain focus on improving the quality of the reasoning path. For example, MCT Self-Refine (MCTSr) integrates a self-correction mechanism directly into the MCTS loop, allowing the LLM to refine its own reasoning steps during exploration Zhang et al. (2024c). Similarly, LLaMA-Berry employs a pairwise preference reward model to globally evaluate and compare different reasoning trajectories, guiding the search toward more promising solutions Zhang et al. (2025c). Other works focus on the efficiency and scalability of the search process. To handle long chains of thought without excessive computational cost, Markov Chain of Thought (MCoT) compresses previous steps into a concise state representation Yang et al. (2024). Some methods circumvent the need for expensive, step-by-step human annotations by training value models on final outcomes alone Yu et al. (2023a) or by using MCTS to automate the collection of process supervision data Luo et al. (2024). To further refine the search, techniques like Constrained MCTS (CMCTS) limit the action space to more rational steps Lin et al. (2025b), while others use lightweight energy functions as path verifiers to guide the search without additional model fine-tuning Xu (2023).

#### E.4.3 CODE GENERATION & SOFTWARE ENGINEERING

In this domain, MCTS is employed to navigate the vast and complex search space of possible code implementations. A significant advantage here is the availability of immediate, objective feedback from external tools like compilers, unit tests, and formal verifiers, which can serve as powerful reward signals. Several works leverage this feedback to guide the search toward correct and efficient code. For instance, RethinkMCTS searches over the reasoning process (i.e., the "thoughts" behind the code) and uses detailed execution feedback to refine erroneous thoughts and steer the search

Li et al. (2024b). Going a step further, VerMCTS generates formally verified programs by using a logical verifier to check the correctness of partial programs at each node in the search tree, providing strong guarantees of soundness Brandfonbrener et al. (2024b). The application of MCTS is broad, spanning from hardware design, where it optimizes for power, performance, and area (PPA) in RTL code DeLorenzo et al. (2024), to complex, repository-level software engineering tasks. In these larger-scale scenarios, multi-agent frameworks like SWE-Search and SWE-Debate use MCTS to manage self-improvement mechanisms and coordinate patch generation Antoniades et al. (2024); Li et al. (2025b). Beyond code generation, MCTS is also used for automated program repair (APRM-CTS) Hu et al. (2025a) and even for evaluating code correctness in an LLM-as-a-Judge paradigm (MCTS-Judge) Wang et al. (2025c). These methods often improve performance by searching over abstract plans rather than raw code, which helps generate more diverse and effective solutions Wang et al. (2024d).

### E.4.4 LLM AGENTS & INTERACTIVE ENVIRONMENTS

For LLM agents operating in interactive environments, where a sequence of decisions is required to achieve a goal, MCTS provides a principled planning mechanism to explore possible action trajectories. These agents must navigate dynamic states and often rely on environmental feedback to guide their choices. A common approach is to use the LLM itself as both a world model to predict future states and a policy to suggest promising actions, effectively combining the LLM's common-sense knowledge with the structured exploration of MCTS Zhao et al. (2023); Yu et al. (2023b). This paradigm has been successfully applied to complex web navigation tasks, where tree search allows agents to perform explicit exploration and multi-step planning, significantly improving success rates on benchmarks like VisualWebArena and WebArena Koh et al. (2024); Zhang et al. (2025f). To manage the immense search space, some frameworks use learned world models to create simulated environments for efficient planning Gao et al. (2025) or leverage learned skills to prune the action space Xie et al. (2025). The versatility of MCTS also extends to specialized domains such as automated machine learning (AutoML), where agents like SELA explore different pipeline configurations Chi et al. (2024), and conversational agents, where MCTS helps plan dialogue actions to ensure conversations are both goal-oriented and compliant with predefined procedures Li et al. (2024c). These frameworks, like Language Agent Tree Search (LATS), unify reasoning, acting, and planning, often incorporating self-reflection to enhance decision-making Zhou et al. (2023).

### E.4.5 RETRIEVAL-AUGMENTED GENERATION (RAG) & KNOWLEDGE-INTENSIVE TASKS

In knowledge-intensive tasks, MCTS enhances RAG by transforming the typically static, one-shot retrieval process into a dynamic and iterative reasoning loop. Instead of retrieving all necessary information at the beginning, MCTS-based approaches strategically decide when to query an external knowledge source and what to ask for at each step of the reasoning process. This allows the LLM to build a solution incrementally, using retrieved information to verify facts, fill knowledge gaps, and correct its trajectory. Frameworks like SearChain and RAG-Star explicitly model this process, using MCTS to explore a tree of reasoning steps where each node can trigger a retrieval and verification action Xu et al. (2024b); Jiang et al. (2024). This dynamic integration of retrieval and reasoning is crucial for mitigating hallucinations and improving factual accuracy, especially in complex multi-hop question answering Wu et al. (2023); Choi et al. (2023). The search can be structured to navigate complex knowledge bases Luo et al. (2025); Xiong et al. (2025); Huang et al. (2024) or to select an optimal combination of retrieved text chunks to feed into the LLM's context Wang et al. (2024i). Some innovative approaches, like FREESON, even empower the LLM to perform the retrieval itself by traversing the corpus using a specialized MCTS, eliminating the need for a separate retriever model Kim & Kim (2025). This tight coupling of search and retrieval enhances the deliberative reasoning capabilities of LLMs, allowing smaller models to tackle complex knowledge-intensive tasks effectively Hu et al. (2025b); Dou et al. (2025).

### E.4.6 MULTIMODAL REASONING

For tasks that require reasoning over both text and other modalities like images or video, MCTS serves as a powerful tool to explore the complex interplay between different data types. It helps to structure the reasoning process by breaking down a multimodal problem into a sequence of steps, where each step can involve grounding textual concepts in visual evidence. For example, the AR-

MCTS framework uses an active retrieval mechanism within the MCTS loop to fetch relevant supporting insights from a hybrid-modal corpus at each reasoning step, ensuring that the generated explanation is well-supported by both visual and textual facts Dong et al. (2024). Other approaches, such as AStar, leverage MCTS in a training-free manner to first abstract a library of high-level reasoning patterns, or "thought cards", from a small set of example problems. During inference, the most relevant thought card is retrieved to provide a strategic scaffold for solving a new multimodal problem, guiding the model's reasoning process without requiring extensive fine-tuning Wu et al. (2025a). Some works also explore using multiple models in a collaborative MCTS framework to jointly search for the best reasoning path, leveraging collective intelligence to tackle difficult multimodal questions Yao et al. (2024a). By systematically exploring how to combine and re-rank multimodal reasoning contexts, these methods make vision-language models more robust and capable of handling complex, multi-step visual reasoning Yang et al. (2025a).

### E.5 MCTS for Self-Improvement via Data Generation

This powerful paradigm uses MCTS not just to find a single good answer, but to generate high-quality reasoning trajectories. These trajectories are then used as synthetic data to fine-tune the LLM or a reward model, creating a virtuous cycle of self-improvement.

### E.5.1 Foundational Self-Improvement Frameworks

These papers introduce the core methodologies for using MCTS within a self-training loop, often inspired by reinforcement learning concepts like AlphaZero and preference optimization. A central theme is the creation of a self-evolutionary cycle where a policy model (the LLM) and a value/reward model are iteratively improved. For example, frameworks like rStar-Math and AlphaLLM use MCTS to perform extensive rollouts, generating vast amounts of verified, step-by-step reasoning data that is then used to train both the LLM and a process preference model Guan et al. (2025); Tian et al. (2024b). This AlphaZero-like approach, where the model learns from its own planned-out explorations, can be adapted to various tasks and model sizes, leveraging a learned value function to guide the search more effectively than relying on a pretrained LLM's priors alone Feng et al. (2023). The data generated from MCTS rollouts is often formatted into preference pairs (i.e., comparing a better reasoning step to a worse one) and used with algorithms like Direct Preference Optimization (DPO) to update the model's policy Xie et al. (2024b); Chen et al. (2024b). This process can be entirely self-contained, as demonstrated by frameworks like AlphaMath, which automatically generate both process supervision and step-level evaluation signals without any human or superior-model annotations Chen et al. (2024a). These methods often focus on learning from both successful and unsuccessful trajectories to enhance generalization Putta et al. (2024); Yuan et al. (2025) and use the search process to explicitly find and correct errors, thereby teaching the model robust recovery skills Kim et al. (2025); Wang et al. (2024g).

### E.5.2 General Capabilities & Alignment

MCTS is used to generate synthetic data for enhancing core LLM capabilities and ensuring alignment with human values. This includes optimizing prompts, where frameworks like PromptAgent treat prompt engineering as a strategic planning problem and use MCTS to explore the space of possible instructions, learning from errors to generate expert-level prompts Wang et al. (2023a). A similar search-based optimization can be used for tuning-free self-alignment, crafting optimal alignment instructions at inference time without costly model updates Singla et al. (2024). In the context of safety, MCTS can generate step-level reasoning data to teach models how to identify and mitigate risks, balancing helpfulness and harmlessness Zhang et al. (2025g). The data generation process can also be used for instruction tuning, where MCTS helps explore the "evolutionary space" of instructions to synthesize high-quality, diverse, and complex training data Li et al. (2024a). By generating data from MCTS trajectories that include both successes and recoveries from failure, models can be trained to be more robust and reflective agents Yu et al. (2024). Some methods guide generation with a discriminator to ensure outputs adhere to constraints like non-toxicity Chaffin et al. (2021), while others leverage the value model from a prior alignment process (like PPO) to guide the search Liu et al. (2023); Khanov et al. (2024).

### E.5.3 Scientific & Specialized Domains

The self-improvement paradigm is being adapted to a wide array of specialized domains. This includes generating high-quality synthetic tabular data Locowic et al. (2024), creating data for multi-agent collaboration Ye et al. (2024), and developing domain-specific models through self-evolution, such as for clinical reasoning in medicine Jiang et al. (2025b). In conversational AI, MCTS-generated dialogue plans are used to train more strategic and effective recommender agents Du et al. (2024). The approach is also used at a meta-level, for tasks like discovering optimal heuristics for optimization problems Zheng et al. (2025) or even optimizing hyperparameters for fine-tuning Volkova et al. (2024). In strategic domains like game-playing, MCTS guides the learning of high-level strategies through self-play simulations Guo et al. (2024); Light et al. (2024). While some applications use MCTS strictly for test-time guidance in specialized areas like therapeutic peptide generation Tang et al. (2025) or catalyst design Sprueill et al. (2023), the broader trend is to use the explored trajectories to create a feedback loop that continually improves the model's domain-specific expertise. Similarly, in molecular structure elucidation, K-MSE Zhuang et al. (2025) leverages MCTS to enhance LLMs with a knowledge base and a molecule-spectrum scorer, significantly improving their chemical reasoning capabilities. This is also seen in multilingual translation, where MCTS is used to generate synthetic data without parallel corpora Zou et al. (2025), and in educational applications for generating personalized test questions Wu et al. (2025c).

### E.5.4 Multimodal Applications

The data generation paradigm extends to multimodal contexts, where MCTS is used to enhance the reasoning capabilities of Vision-Language Models (VLMs). To overcome the lack of fine-grained supervision in multimodal reasoning, MCTS-based pipelines can automatically generate millions of step-level annotations for training powerful process reward models (PRMs) without human labeling Du et al. (2025). Another approach involves creating a multimodal actor-critic framework where MCTS guides an actor model to explore diverse reasoning paths. An annotator model then compares pairs of paths-one leading to a correct outcome and one to an incorrect one-to generate critique data that teaches the VLM to correct its own errors Liu et al. (2025b). An alternative, data-efficient strategy uses MCTS to quantify the difficulty of visual reasoning samples by measuring the number of search iterations required to solve them. This allows for the selection of a small but highly informative subset of challenging examples for reinforcement fine-tuning, achieving state-of-the-art performance with significantly less data Wang et al. (2025b).

## F    Informed Search Based Method

To enhance the reasoning capabilities of Large Language Models beyond simple sequential generation, researchers have increasingly turned to informed search algorithms. This paradigm structures problem-solving as a tree traversal, where heuristic guidance helps navigate vast and complex solution spaces efficiently. Early frameworks such as Tree-of-Thoughts (ToT) adapted classical algorithms like Breadth-First Search (BFS) and Depth-First Search (DFS), using the LLM itself to evaluate intermediate 'thoughts' and prioritize promising reasoning paths. Building on this, more recent approaches have implemented A* search, a more sophisticated heuristic method, to further optimize exploration. Methods like ToolChain* and Q* exemplify this trend by designing intricate cost and heuristic functions that incorporate memory, self-consistency, and learned value estimates to guide the search for optimal solutions. This section explores these key informed search strategies, detailing how they formalize and direct the LLM's reasoning process.

### F.1    Informed BFS/DFS

The Tree-of-Thoughts (ToT) framework (Yao et al., 2024b) enables Large Language Models (LMs) to systematically explore multiple reasoning paths. It formulates problem-solving as a tree search, where each node is a state $s = [x, z_{1...i}]$ comprising the input $x$ and a sequence of thoughts $z_{1...i}$ generated thus far. The ToT framework is defined by four key components: problem structuring, thought generation, state evaluation, and a search strategy.

The framework first **decomposes** the problem into intermediate steps. Then, at each step $i + 1$, a generator $G(p_\theta, s, k)$ produces $k$ candidate thoughts from a given state $s = [x, z_{1...i}]$ using an LM

$p_\theta$. This generation occurs via two distinct methods: (1) **sampling** $k$ independent and identically distributed (i.i.d.) thoughts from a Chain-of-Thought (CoT) prompt, a method effective for expansive thought spaces (e.g., text generation); or (2) **proposing** thoughts sequentially using a "propose prompt" to prevent redundancy, which is better suited for constrained reasoning tasks. To guide the search, an evaluation function $V(p_\theta, S)$ leverages an LM $p_\theta$ to provide heuristic assessments of progress for a set of states $S$. The evaluation can be performed in two modes: (1) a **value-based** approach, where each state is scored independently, yielding a scalar or categorical assessment; or (2) a **voting-based** approach, where the LM selects the most promising state from the set $S$.

ToT implements two primary search algorithms. The **informed Breadth-First Search (BFS)** algorithm emulates a beam search, maintaining a beam of $b$ states at each step. This process constrains the number of states at any depth to $b$, avoiding exponential growth and making it efficient for problems with a fixed depth $T$. In contrast, the **informed Depth-First Search (DFS)** algorithm explores a single path until its value, as determined by the evaluator, falls below a threshold, at which point the path is pruned.

Building on these foundational search strategies, recent works have adapted BFS-style exploration for a variety of specialized domains. In causal discovery, LLM-guided BFS has been employed to efficiently uncover causal graphs from both textual knowledge and observational data, using dynamic scoring and active learning to navigate the hypothesis space (Jiralerspong et al., 2024; Susanti & Färber, 2025; Zanna & Sano, 2025). Beyond structured discovery, researchers have also explored the LLM's intrinsic capacity for search. For instance, the Autonomous Tree-Search (ATS) paradigm demonstrates that LLMs can execute a BFS-like exploration internally with a fixed system prompt, eliminating the need for external control logic (Zhang et al., 2023b). Other work has proposed LLM-First Search (LFS), where the model itself dynamically decides whether to broaden the search (go wider) or deepen the current path, offering a more adaptive alternative to the fixed beam width of ToT-BFS (Herr et al., 2025). In more fundamental architectural explorations, a novel paradigm called Coconut (Chain of Continuous Thought) has shown that by reasoning in a continuous latent space, LLMs can implicitly perform BFS to explore multiple reasoning steps simultaneously (Hao et al., 2024). For highly structured domains like automated theorem proving, BFS-Prover integrates Best-First Search with an expert iteration framework, achieving state-of-the-art results by strategically filtering problems and refining its policy with Direct Preference Optimization (DPO) (Xin et al., 2025).

### F.2 A*

To mitigate the computational overhead associated with methods like Monte Carlo Tree Search (MCTS), recent work has explored A*-based search algorithms. These methods have been particularly prominent in robotics, where frameworks like LLM-A* leverage the commonsense knowledge of LLMs to generate heuristics for path planning, synergizing the precise pathfinding of A* with the global reasoning of LLMs (Meng et al., 2024). Notably, ToolChain* (Zhuang et al., 2023) and Q* (Wang et al., 2024b) apply A* search at inference time for general reasoning tasks.

These methods guide exploration using a specialized cost function $f(n) = g(n) + h(n)$, which prioritizes nodes that appear to be on the most promising path to a solution. This function balances the cost of the path taken so far, $g(n)$, with an estimated cost to reach the goal, $h(n)$. The primary innovation in methods like ToolChain* (Zhuang et al., 2024) and Q* (Wang et al., 2024c) lies in constructing composite heuristics for $g(n)$ and $h(n)$ from diverse, LLM-relevant signals. The key components used to formulate these cost functions are summarized in Table 6. The key components used to formulate these cost functions are summarized in Table 6.

**ToolChain***

In ToolChain*, the cost function for a node $n$ is the standard A* formulation, $f(n) = g(n) + h(n)$, where $g(n)$ is the **cumulative cost** from the start node to $n$, and $h(n)$ is a heuristic estimate of the **future cost** to the goal. The cumulative cost $g(n)$ is the sum of single-step costs over all ancestors of $n$, denoted $an(n)$. Each single-step cost is derived from two value functions, $g_{t,1}$ and $g_{t,2}$, whose outputs are bounded in $[0, 1]$. The cost is formulated as the geometric mean of the complements of

Table 6: Compact Overview of A* Heuristic Components for LLMs

| Heuristic | A* Component | Mechanism (and Signal Source) |
|---|---|---|
| Process-Based Rewards | $g(n)$ | Aggregates step-wise rewards from execution feedback (e.g., logits, rule checks). |
| Statistical Consistency | $g(n)$ | Favors steps that are frequently proposed across multiple generation samples. |
| Memory-Based Comparison | $g(n), h(n)$ | Scores path similarity against a repository of high-quality examples (e.g., using LCS). |
| Learned Future Value | $h(n)$ | Estimates the cost-to-goal using a trained proxy model (e.g., a Q-function). |

these values. The cumulative cost is thus:

$$g(n) = \sum_{i \in \{an(n), n\}} (1 - g_{t,1}(i))^{\alpha} \cdot (1 - g_{t,2}(i))^{1-\alpha}, \tag{6}$$

where the hyperparameter $\alpha$ weights the contribution of each value function.

The first value function, $g_{t,1}(n)$, is task-specific and draws from a **long-term memory** $\mathcal{M}$, which is initialized with seed demonstrations and augmented with successful plans discovered during search. Each memory entry $m_j$ is a plan sequence $(s_{j,0}, a_{j,1}, \ldots, a_{j,T_j})$. This function evaluates the current plan $s_n$ by computing its maximum longest common subsequence (LCS) score against all plans in memory: $g_{t,1}(n) = \max_{m_j \in \mathcal{M}} \frac{\text{LCS}(s_n, m_j)}{\min(L(s_n), L(m_j))}$, where $L$ is the sequence length. The second value function, $g_{t,2}(n)$, is based on **self-consistency frequency**. It measures the frequency with which node $n$ is proposed as the next step across $k$ independently sampled reasoning paths, reflecting its reliability.

The **future cost** $h(n)$ is formulated analogously to $g(n)$:

$$h(n) = \sum_{i \in \{an(n), n\}} (1 - h_{t,1}(i))^{\beta} \cdot (1 - h_{t,2}(i))^{1-\beta}, \tag{7}$$

where $\beta$ is the geometric mean weight. The first heuristic, $h_{t,1}(n)$, leverages the **long-term memory** $\mathcal{M}$. For an action node $n$, it finds the action $a$ in each memory plan $m_j$ with the highest lexical similarity to $n$. The heuristic is the sum of these actions' relative positions: $h_{t,1}(n) = \sum_{m_j \in \mathcal{M}} \mathbf{1}_{a \in m_j} \frac{pos(a, m_j)}{T_j}$. The second heuristic, $h_{t,2}(n)$, is an **LLM imagination score**. An LLM generates a plausible future plan toward a target node $n_T$, and the heuristic value is the ratio of the current path length to the total imagined path length: $h_{t,2}(n) = \frac{|an(n)|}{|an(n_T)|}$, where $|an(\cdot)|$ is the number of ancestors. A higher score signifies closer proximity to the goal.

### Q*

In Q*, the cost function is $f(n) = g(n) + \lambda h(n)$, where $\lambda$ is a weighting hyperparameter. The accumulated cost $g(n)$ is an aggregation of process-based rewards for the current node and its ancestors: $g(n) = \text{Agg}(\{\mathcal{R}(s) \mid s \in an(n) \cup \{n\}\})$. The reward function $\mathcal{R}$ can be derived from human feedback, ground-truth labels, predefined rules, or LM logit scores. The aggregation function, Agg, can be chosen from $\{\max, \min, \sum, [-1]\}$, where $[-1]$ indicates selecting the reward of the last node.

The heuristic cost $h(n)$ is a Q-function that estimates the expected future reward. As an exhaustive search over subsequent steps is intractable, the heuristic is approximated by taking the maximum Q-value among the top-$k$ actions proposed by the LLM policy $\pi_\theta$: $h(n) = \max_{a_t \in \text{top-k}(\pi_\theta(\cdot|n))} Q(n, a_t)$. A primary challenge is estimating optimal Q-values when the frozen policy $\pi_\theta$ is suboptimal. The authors propose three methods for learning a proxy Q-value model: (1) offline reinforcement learning on curated data, (2) learning from MCTS rollouts, or (3) distillation from a stronger LLM. However, this approach may have limited generalization, and the anticipated computational savings are not guaranteed.

# G  UNIFIED EVALUATION AND COMPUTE ACCOUNTING FOR TREE-SEARCH

To characterize the current capabilities of Tree-Search Test-Time Scaling (TTS), we select **mathematical reasoning** as the representative domain. We prioritize this domain because, unlike open-ended generation, mathematical problems offer deterministic success criteria, enabling high-resolution analysis. While our case study focuses on GSM8K and MATH, the fragmentation it reveals in reporting and compute accounting is systemic (Kaplan et al., 2020b; Hoffmann et al., 2022b; Snell et al., 2024). Consequently, the framework we propose here is deliberately **domain-agnostic** and intended as a reusable standard.

## G.1  THE LANDSCAPE OF MATHEMATICAL REASONING AND THE INFEASIBILITY OF RETROSPECTIVE COMPARISON

To concretely visualize the state of the art, we examine canonical benchmarks where tree-structured decoding has shown substantial gains (Xie et al., 2024a; Ha et al., 2025; Guan et al., 2025). As visualized in Figure 7, MCTS-based variants like MCTSr and rStar-Math populate a Pareto frontier that dominates standard baselines, reinforcing a form of *model–search equivalence* where smaller models with search rival larger static models.

However, we emphasize that **a strictly fair, compute-normalized comparison of existing literature is currently infeasible**. Unlike controlled studies (Snell et al., 2024), published tree-search papers exhibit substantial methodological heterogeneity that prevents retrospective normalization. **First**, verifier costs are frequently opaque; many methods employ deep neural Reward Models without reporting the associated token overhead ($T_{\text{eval}}$), making it impossible to calculate total FLOPs without original logs. **Second**, hardware platforms diverge significantly (e.g., A100 clusters vs. consumer GPUs), rendering wall-clock comparisons invalid. **Third**, baselines span massive parameter scales ($\sim$7B to 70B+), preventing simple step-based comparisons.

**Logical Implication:** Constructing a truly apples-to-apples ranking under a unified protocol would require re-implementing and re-evaluating all surveyed methods from scratch. Such an undertaking constitutes a comprehensive *benchmarking study* in its own right, distinct from the scope of this *methodological survey*. Therefore, rather than attempting an imprecise retrofit of past results, we propose a forward-looking protocol to resolve this fragmentation in future work.

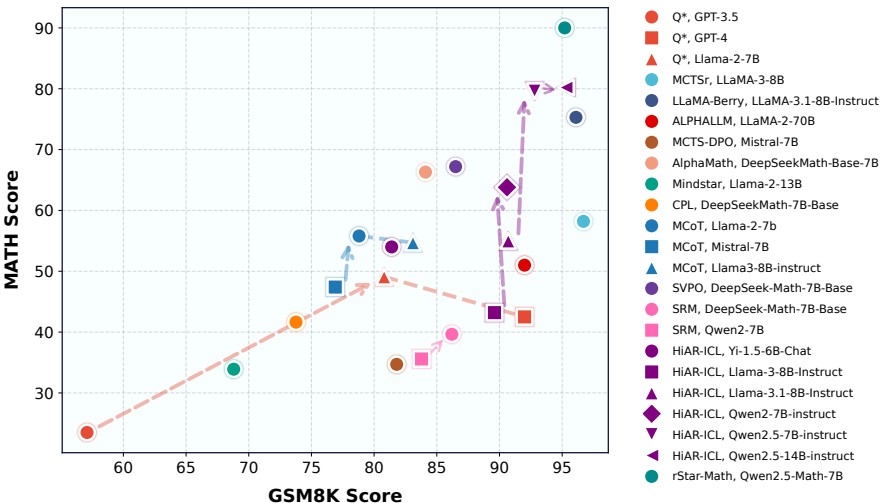

Figure 7: Performance landscape of tree-search methods across GSM8K and MATH. **Caveat:** The scatter plot aggregates reported metrics from heterogeneous experimental setups. Due to missing data on verifier costs and unstandardized compute budgets in the original papers, **re-computing these data points under a unified FLOPs standard is impossible**. This visualization conveys the qualitative state-of-the-art rather than a controlled iso-compute ranking.

### G.2 PROPOSED PROTOCOL: A UNIVERSAL FRAMEWORK FOR COMPUTE ACCOUNTING (SCRP)

To address the systemic issues identified above, we propose the **Standardized Compute-Reporting Protocol (SCRP)**. This protocol provides a minimal, actionable recipe for comparability without requiring retroactive adjustments to baseline data.

**Unified Resource Vector and FLOPs Abstraction.** We first disentangle compute sources by defining a budget vector $\mathbf{B} = (C_{\text{policy}}, C_{\text{eval}}, C_{\text{verify}}, T_{\text{wall}})$, which explicitly separates policy expansion, node scoring, and external verification. To normalize across heterogeneous hardware, we advocate using FLOPs as the primary independent variable. Specifically, for a $P$-parameter dense transformer, we approximate the inference cost as $C \approx 2 \cdot P \cdot T$. The total compute cost for an instance $x$ aggregates all components:

$$\mathcal{C}_{\text{total}}(x) \approx \underbrace{2 \cdot P_{\text{policy}} \cdot T_{\text{policy}}(x)}_{\text{generation}} + \underbrace{2 \cdot P_{\text{eval}} \cdot T_{\text{eval}}(x)}_{\text{evaluation}} + C_{\text{verify}}(x) \qquad (8)$$

where $T_{\text{policy}}$ and $T_{\text{eval}}$ track the cumulative tokens generated and processed, and $C_{\text{verify}}$ accounts for symbolic execution costs.

**Standardized Metrics.** Based on this budget, we recommend reporting three key metrics: (1) **Budgeted Accuracy (Pass@FLOPs)**, defined as $Q(b) = \mathbb{E}[\text{Acc} \mid \mathcal{C}_{\text{total}} \leq b]$, which explicitly visualizes the trade-off between search depth and accuracy; (2) **Tokens-per-Solved (TpS)**, a model-agnostic proxy for search algorithm efficiency; and (3) **Parallelism Efficiency**, the ratio between theoretical FLOPs and realized wall-clock speedup. Adopting SCRP allows future research to produce naturally comparable compute-performance curves, eliminating the opacity that currently plagues the field.

## H CHALLENGES AND FUTURE OF TREE-SEARCH METHODS

**Search Efficiency and Intelligence**. Tree search algorithms, despite their power, often require significantly greater computational resources than greedy decoding, as noted by Wang et al. (2024a), with resource demands exceeding 10 times that of greedy approaches in certain cases due to inefficiencies in search strategies. This high computational overhead presents a substantial barrier to the practical deployment of these methods. Algorithms like MCTSr and LLaMA-Berry, which generate multiple solutions sequentially at each node, exacerbate these resource demands. To mitigate these limitations, future research could prioritize improving the efficiency of tree search algorithms by investigating trade-offs between policy and reward models, incorporating dynamic control mechanisms, and employing effective pruning techniques to optimize tree expansion.

**Overthinking Issues in Simple Queries.** Task complexity is closely related to the length of reasoning chains, highlighting the need for extended cognitive processing in more difficult problems (Qin et al., 2024; Huang et al., 2025). However, Chen et al. (2024d) and Zeng et al. (2024a) observe that O1-like models often overanalyze simple questions, dedicating excessive computational resources to tasks that have clear and obvious answers. For instance, a query like "3-2=?" does not require complex reasoning, yet these models may engage in unnecessary computations, wasting resources and potentially introducing errors. Forcing models to reason through such trivial tasks not only consumes valuable computational power but also causes delays. Future research should focus on methods to reduce these inefficiencies, improving models' ability to quickly recognize and handle straightforward queries while dynamically allocating computational resources across diverse problem types.

**Self-play Between Policy Models and Reward Models.** Certain tree-search algorithms encounter challenges due to limited parallelism, which constrains their search speed, especially in resource-intensive settings. As detailed in Section E, various tree-search techniques can generate traces that are then employed to iteratively refine reward and policy models, such as ReST-MCTS and rStar-Math. This self-play paradigm is crucial for internalizing the reasoning system into the policy model, thereby endowing LLMs with sophisticated reasoning abilities (Xiang et al., 2025). By internalizing tree-search reasoning into LLMs, the tree-search process can be structured within a CoT framework, facilitating sequential reasoning. This not only enhances reasoning efficiency but also mitigates

parallelism limitations, thereby improving scalability. Future research should investigate strategies to optimize this self-play paradigm further, facilitating more efficient problem-solving.

**Reward Modeling and Reward Model Training.** Section E examines various MCTS-based evaluation strategies. A central element of the search strategies is the reward or evaluation model, which provides essential supervision to guide search processes effectively (Lightman et al., 2023; Setlur et al., 2024; Xiang et al., 2025). Reward models are broadly categorized into two types: the Outcome Reward Model (ORM) and the Process Reward Model (PRM). Unlike outcome rewards, which deliver feedback only at the task's conclusion, process rewards provide signals at both intermediate steps and the final outcome, enabling finer-grained and more frequent supervision. Nevertheless, learning process rewards present significant challenges. For example, Uesato et al. (2022); Lightman et al. (2023) relies on human annotators for process supervision, a costly and inherently unscalable method. While automated methods for constructing process rewards have been proposed (Wang et al., 2024e; Luo et al., 2024; Wang et al., 2024h), they are predominantly designed for specialized areas such as mathematics and programming. These approaches struggle to generalize to broader domains, such as scientific reasoning and complex problem-solving, where human evaluation remains essential. Overcoming these limitations necessitates the development of more efficient methods to generate high-quality fine-grained rewards and scalable techniques to advance reward model capabilities, which remain open and pressing research challenges.

**Reward Model Quality and Its Effect on Search.** The performance and efficiency of search during testing depend on the quality of the Process Reward Model (PRM)(Setlur et al., 2024; Xiang et al., 2025). However, searches guided by an oracle verifier are more efficient than those relying on a learned PRM(Anonymous, 2024). Numerous studies have shown that an imperfect reward model can give rise to inverse inference scaling (Zeng et al., 2024b). For instance, Gao et al. (2023) identified an inverse scaling effect, where expanding the search space in best-of-n search negatively impacts performance due to a distribution shift between the imperfect reward model and the policy model. These findings underscore the critical need to bridge the performance gap between oracle and learned reward models. Xiang et al. (2025) shows that while the PRM's ability to verify complete solutions improves with additional data, a notable gap persists between trained PRMs and oracle PRMs. Therefore, understanding how scaling laws for process supervision models influence their effectiveness and efficiency in large-scale search tasks remains a pivotal challenge.

# I   THE USE OF LARGE LANGUAGE MODELS (LLMS)

Large Language Models (LLMs) were used as assistive tools in the preparation of this work. Specifically, we employed GPT-5 to make minor edits to academic writing, such as drafting and refining sections. All scientific claims, methodological contributions, and experimental results were conceived, implemented, and validated by the authors. The authors take full responsibility for the content presented in this paper.

