# OpenReview forum: "A Unified Perspective and Review on Tree Search for LLMs Test-Time Scaling"
_ICLR.cc/2026/Conference — ICLR 2026 Conference Withdrawn Submission_

### Official Review · Reviewer_yddj · 2025-10-30

**Soundness:** 3
**Presentation:** 2
**Contribution:** 3
**Rating:** 4
**Confidence:** 4

**Summary:**

This paper presents a comprehensive, systematized review of tree search–based methods for test-time scaling (TTS) in LLMs. From a unified mathematical perspective, it clearly articulates the operating mechanisms and core design choices of different tree-search algorithms, with a particular emphasis on reward formulation. Within the MCTS family, it contrasts methods along key axes—node representation granularity, process- vs. outcome-based rewards, and external evaluators vs. self-evaluation—to delineate strengths, and limitations. The survey also synthesizes major applications of these techniques and outlines current challenges together with promising future directions.

**Strengths:**

1. **Comprehensive and unified framing.** The paper surveys tree-search methods for TTS—from classical DFS/BFS through heuristic search to MCTS—and proposes a unified framework that foregrounds core design choices and reward formulations, providing a clear basis for cross-method comparison.

2. **Well-chosen design axes with comprehensive and detailed coverage.** The review systematically contrasts, for example, process- vs. outcome-based rewards, node-granularity choices, and external evaluators vs. self-evaluation, and explains their implications for applicability, strengths, and limitations.

3. **High reference value and readability.** The unified notation, consistent taxonomy, and well-structured exposition make the survey a useful reference for practitioners and an accessible entry point for newcomers.

**Weaknesses:**

1. **Lack of a clear, logical hierarchy.** While the main text is reasonably readable given the page limits, the appendices—though comprehensive—are organized in a strictly top-down manner that hurts skimmability and makes it hard to quickly locate the key takeaways. For example, in Appendix D: MONTE CARLO TREE SEARCH (MCTS), the authors first classify algorithms along *Tree Node*, *Node Evaluation*, and *Evaluation Need*, and then introduce 15 algorithms sequentially. The length and ordering make it difficult to grasp each algorithm’s core ideas, overlapping applicability, and points of comparison. A more hierarchical, comparative presentation (e.g., taxonomy tables or side-by-side comparisons designed from a reader’s perspective) would improve readability.

2. **Potential drift from the central focus.** I question the value of the section comparing reward in search vs. reward in RL. The paper’s stated focus is tree search for LLM test-time scaling. Rewards in search and in RL are inherently different constructs; their distinction is fairly straightforward. It is not clear why this comparison is included here or how it advances the paper’s central argument.

3. **Typos and minor errors.** For instance:
   Section 2.2, paragraph 2, line 4: “not correctness..”,
   Section 2.3, paragraph 2, line 3: “has been taken from s. ,”.
   Please proofread and standardize wording and punctuation throughout.

**Questions:**

1. **Task-oriented MCTS guide.** Could you provide a task-based comparison and practitioner’s guide to MCTS variants—mapping representative tasks (e.g., math, code, RAG, agents) to recommended node granularity (trace / state–action / terminal), evaluation signal (PRM / ORM / hybrid), selection/backup rules, and typical hyperparameter ranges?

2. **Standardized MCTS benchmarking.** Do you plan to present comparisons of MCTS methods under a standardized protocol—covering datasets, fixed token budgets, degree of parallelism and shared metrics (e.g., *Budgeted Accuracy*, *Tokens-per-Solved*) ?

---

> ### Author Response · Authors · 2025-11-21
> **Official Comment by Authors**
>
> We thank all reviewers for their thoughtful and constructive feedback. Your comments helped us clarify the conceptual framing, improve the structure of the survey, and strengthen the practical guidance provided to readers. `We have revised the manuscript accordingly, with all additions and edits highlighted in blue in the updated PDF.`
>
> #### **Summary of Revisions**
>
> In this revision, we have made substantial improvements to both the structure and clarity of the survey. The key changes are:
>
> 1. **Unified and Reorganized Appendix Structure:**
>    We restructured the entire appendix into a coherent hierarchy (Modules A–H), introduced an Appendix A roadmap, and reorganized methodological details into clearer taxonomies for both MCTS and heuristic search.
>
> 2. **Clarified Reward Distinction (Section 2.4):**
>    We refined the explanation of *search-time reward* vs. *learning-time reward* and strengthened the conceptual bridge needed to understand hybrid methods such as ReST-MCTS\* and AlphaLLM.
>
> 3. **Expanded MCTS Design Taxonomy (Section 3):**
>    We added clearer definitions for node representation, evaluation locus, multi-critic designs, and planning-based state transitions, supported by citations to recent LLM planning frameworks.
>
> 4. **Added a Practitioner-Oriented MCTS Guide:**
>    We introduced a  expanded **Table 2 in the main text** and new **Appendix E.2** to provide task-specific configurations across Math, Code, RAG, and Agentic environments.
>
> 5. **Standardized Compute Reporting (Appendix G):**
>    We added the SCRP (Standardized Compute-Reporting Protocol), a unified FLOPs-based accounting framework, and introduced new benchmark metrics such as **Budgeted Accuracy (Pass@FLOPs)** and **Tokens-per-Solved (TpS)**.
>
> 6. **Proofreading and Minor Corrections:**
>    All noted typos, punctuation issues, and notation inconsistencies have been corrected.
>
> `These revisions collectively improve the readability, technical precision, and practical value of the survey.`

---

> ### Author Response · Authors · 2025-11-21
> **Official Comment by Authors**
>
> > *R1: Lack of a clear, logical hierarchy.*
>
> In the revision, we have substantially **restructured the entire appendix** into a coherent hierarchy, now organized as **six modules** (summarized in the new **Appendix A**):
>
> 1. **Foundational Paradigms (App. B)** — Revisits classical search (DFS/BFS/A*) to unify terminology.
> 2. **Theoretical Distinctions (App. C & D)** — Formalizes Test-Time Scaling and clarifies reward-as-learning vs reward-as-planning.
> 3. **MCTS Methodological Taxonomy (App. E)** — Structured breakdown of node granularity, evaluation locus, and backup rules.
> 4. **Informed Search Taxonomy (App. F)** — Detailed analysis of LLM-generated heuristic search (Tree-of-Thoughts, ToolChain*, Q*).
> 5. **Standardized Evaluation (App. G)** — Unified compute protocol.
> 6. **Future Directions (App. H)** — Open challenges such as reward fidelity and efficiency bottlenecks.
>
> We further added a concise roadmap (`“Organization of the Appendix”`) at the beginning of **Appendix A**, helping readers navigate the layered structure.
>
> ---
>
> > *R2: Potential drift from the central focus..*
>
> We appreciate this comment and have clarified the motivation for Section 2.4. We also refined the section to be more concise and removed peripheral explanations that were not directly necessary for understanding the planning–learning distinction.
>
> **1. Why this distinction is important**
> Terms such as *reward*, *value*, and *policy* are used inconsistently across both RL and MCTS literature. Because this survey aims to unify the space, distinguishing:
> - **Search Reward** (instance-level, transient, inference-time), and
> - **RL Reward** (dataset-level, durable, parameter-updating)
>
> is critical for avoiding conceptual confusion, especially for readers new to tree-search-based TTS.
>
> **2. Why it is structurally necessary**
> Modern hybrid methods (e.g., **ReST-MCTS\***, **AlphaLLM**) intertwine:
> - **search-time planning signals**, and
> - **learning-time updates** (e.g., preference optimization or value-model training).
>
> Section 2.4 now makes this bridge explicit: search reward guides planning for a single instance, while **RL-style updates** adjust model parameters across many episodes. This conceptual connection is required for understanding **Section 3.6.2** and **Appendix D**, which analyze hybrid self-improvement loops.
>
> The revised text in Section 2.4 now states this explicitly and ties it directly to the patterns seen in modern MCTS-enhanced LLMs.
>
> ---
>
> > *R3: Typos and minor errors*
>
> Thank you for the careful reading. We have corrected the specific errors noted (e.g., punctuation in citation groups, clarifying notation definitions) and conducted a thorough proofreading pass to ensure standard formatting across all new sections.

---

> ### Author Response · Authors · 2025-11-21
> **Official Comment by Authors**
>
> > *R4: Task-oriented MCTS guide.*
>
> We appreciate this suggestion, which highlighted an important gap in the original submission.
> In response, we have added concrete, practitioner-oriented guidance that translates the taxonomy into actionable configurations.
>
> **New Additions:**
>
> - **Appendix E.2 (“Practitioner’s Guide”)** now provides *task-specific configurations* for key domains:
>   - Mathematical reasoning (multi-step, high-variance)
>   - Code generation (binary-verifiable)
>   - RAG and knowledge-intensive tasks (retrieval + reasoning)
>   - Agentic settings (state–action transitions)
>
> - This guide distills the design axes—node topology, reward model choice, backup rule, and evaluator budget—into clear recommendations grounded in the literature.
>
> **Condensed excerpt from Table 2:**
> To make the distinctions more immediately visible, we include a subset of the new table here:
>
> | **Task Domain** | **Recommended Node Topology** | **Preferred Evaluation (PRM / ORM)** | **Backup Rule** | **Typical Hyperparameters** | **Representative Methods** |
> |------------------|--------------------------------|---------------------------------------|------------------|-------------------------------|----------------------------|
> | **Math & Logic** | Trace-based (step or solution level) | PRM / PPRM or self-reflection | **Average / Sum** | `cpuct = 1–4`, Rollouts `16–128`, Depth `8–20` | ReST-MCTS*, rStar-Math, LLaMA-Berry |
> | **Code Generation** | Terminal-state (block / function-level) | ORM + verbal feedback | **Max** | Rollouts `16–64`, `k = 5–50`, Temp `0.6–0.8` | PG-TD, RethinkMCTS |
> | **RAG** | Hierarchical (Retrieve → Reason) | Hybrid PRM + ORM | **Min / AND** | Retrieval `k = 3–10`, Depth `3–5`, ≤10 iterations | RAG-Star |
>
> This directly addresses the reviewer’s request for actionable, domain-specific MCTS design guidance.
> We believe this substantially improves the utility of the survey for practitioners.
>
> ---
>
> > **R5: Standardized MCTS benchmarking.**
>
> We appreciate this important question. A key challenge we faced when preparing the survey is that a fully standardized cross-paper benchmark is currently infeasible due to inconsistent compute reporting, heterogeneous hardware, unreported verifier costs, and different backbone scales. Nevertheless, we have taken substantial steps toward enabling such benchmarking in future work.
>
> To directly address this concern, we added a new and substantial **Section 5 and Appendix G: “Unified Evaluation and Compute Accounting for Tree-Search TTS.”**
>
> - **Figure 7: Landscape of Compute Reporting Confounders**
>   We added **Figure 7** to visualize the performance landscape across GSM8K and MATH using *all available* reported metrics from prior papers.
>    The scatter plot highlights a key challenge we confronted directly:
>
>    - Experimental setups differ widely (model scale, verifiers, hardware).
>    - Verifier and evaluator costs are often **missing or undocumented**.
>    - Search budgets and batching strategies are **not standardized**.
>
>    As noted in the caption, **re-computing these points under a unified FLOPs standard is impossible with the currently available information**, despite our attempt to normalize them.
>    Thus, Figure 7 serves as a qualitative map of the research landscape—not an iso-compute ranking—illustrating exactly why benchmarking is difficult today.
>
>
> - **SCRP (Standardized Compute-Reporting Protocol)**
>   A domain-agnostic framework that decomposes inference cost into a unified resource vector:
>   \[
>   B = (C_{\text{policy}}, C_{\text{eval}}, C_{\text{verify}}, T_{\text{wall}})
>   \]
> - **Operational FLOPs approximation**
>   \[
>   C_{\text{total}} \approx 2\cdot P_{\text{policy}} \cdot T_{\text{policy}} + 2\cdot P_{\text{eval}} \cdot T_{\text{eval}} + C_{\text{verify}}
>   \]
> - **New metrics for comparison**
>   - **Budgeted Accuracy (Pass@FLOPs)**
>   - **Tokens-per-Solved (TpS)**
>
> While a retrospective standardized benchmark is not scientifically reliable today (as illustrated in **Figure 7**), the combination of **SCRP**, standardized metrics, and clear compute accounting provides the community with the foundations needed for **future fair comparisons**.
>
> ---
>
> `We thank the reviewer again for these helpful suggestions. We believe the revisions have significantly elevated the quality and utility of this work.`

---

### Official Review · Reviewer_giMx · 2025-10-31

**Soundness:** 3
**Presentation:** 3
**Contribution:** 2
**Rating:** 6
**Confidence:** 2

**Summary:**

This paper provides a survey on tree search methods for scaling test-time compute with LLMs. The paper begins by providing introductory descriptions of various tree search methods, including uninformed search, informed search and MCTS, as well as a short section contrasting test-time search with RL. The paper then dives deeper into MCTS, explaining first the problem formulation and notation used in the MCTS literature, and then the various components of MCTS (nodes, value functions, evaluation, selection and expansion etc.). Throughout this section, the authors connect MCTS to LLM search and describe the various design choices made by recent papers in the area. The paper then proceeds by describing some modern applications of MCTS for LLMs, particularly on how MCTS is used to scale test-time compute in various LLM applications (e.g. RAG, multimodal reasoning) and to build self-improvement methods. Lastly, the authors provide a brief overview of informed search with LLMs (e.g. Tree-of-Thoughts and A* search).

**Strengths:**

- The main paper is well-written and highly accessible, even for someone without prior experience working on LLM tree search research. I believe it makes for a good introductory read for someone new to the research area. Importantly, the authors do a good job connecting tree search back to other active areas of LLM research (e.g. self-improvement, scaling test-time compute), so it is immediately clear why one should care about this topic.
- The authors scout out a good selection of relevant work and use the algorithms introduced in these works to highlight various aspects of MCTS. This paper provides the reader with a good set of starting works to begin their own in-depth exploration of MCTS.
- The appendix contains more detailed explanations of various algorithms and design choices that were introduced in the main text. I found reading this to be especially enlightening.

**Weaknesses:**

While this work may help researchers less familiar or even unfamiliar with LLM tree search/MCTS literature understand the basics, I believe that good survey papers should be useful not only to this group of readers. Rather, survey papers can also introduce useful novel insights that would be helpful even to active researchers in the area without needing to introduce novel algorithms and artefacts, for example by:
1. Unifying disparate results, evidence and arguments found across many different papers and presenting them in a cohesive and structured way.
2. Conducting meta-analysis on top of existing works to unlock new insights that are absent from the individual papers e.g. pointing out unfilled gaps in the literature, inconsistencies in currently-accepted experimental protocols etc.
3. Conducting small experiments that are missing in prior works that may help us better understand how various methods compare.

Such insights are largely absent in this survey. Take, for example, the section comparing ORMs and PRMs for evaluation. Here the authors merely describe the superficial difference between the two approaches and cite some papers that use these models for MCTS without necessarily describing existing evidence for why and when one might be better than the other. A researcher who has read this survey may now have a refreshed memory of what PRMs/ORMs are and might have an updated list of works to refer to, but gained little additional insight that would be helpful to their own new research in the field.

On the actual content of the paper, I can see several gaps that I think could be worth filling:
- Compute trade-offs: tree search algorithms/MCTS are very expensive to run. I think it could be useful including some discussion on (1) when these algorithms are likely to be useful vs. other, potentially cheaper inference-time algorithms (2) what the main "knobs" that we can turn are for trading compute and accuracy, and which knobs we should turn when.
- "Splitting" text into states: when doing MCTS with LLMs, we need to define what a state or a partial trajectory looks like. Is it a single token? A sentence? A "reasoning step"? I think it could be helpful to discuss the trade-offs associated with various design choices here.
- What are some characteristics of the kinds of problems/environments where MCTS may be most helpful? For example, I would guess that MCTS is more helpful when using LLMs in structured, grid-like game environments than when using LLMs to write emails. Knowing this could be helpful for casting light onto when MCTS/tree search should be used vs. other algorithms.
- What are the trade-offs of MCTS vs. informed tree-search algorithms e.g. tree-of-thoughts?

**Questions:**

Despite my lengthy "Weaknesses" section, I do appreciate the utility of this paper as introductory material for researchers new to this line of research, and I do genuinely believe this to be very impactful. I am also somewhat uncertain on how we should consider the impact of survey papers in the main conference track (is being introductory material enough for an acceptance here, or are we looking for more?). I am therefore currently recommending this for a borderline accept, with a low confidence score. I look forward to engaging more with you and the other reviewers on this matter.

---

> ### Author Response · Authors · 2025-11-21
> **Official Comment by Authors**
>
> We thank the reviewer for the thoughtful and constructive feedback. Your comments directly guided several revisions that strengthen the survey’s analytical depth and practical utility. `We have revised the manuscript accordingly. All newly added or modified content has been incorporated into the updated PDF and is highlighted in blue for clarity.`
>
> #### **Summary of Revisions**
>
> To address your requests for deeper synthesis, clearer design principles, and practitioner-oriented guidance, we made the following focused updates:
>
> 1. **Standardized Compute-Reporting Protocol (SCRP).**
>    Added in Section 5 and Appendix G to provide a unified FLOPs- and token-normalized evaluation framework for tree-search TTS methods.
>
> 2. **Clarified MCTS Formalism.**
>    Section 3.1 and Table 1 now give a clearer planning-based interpretation (state/action definitions, deterministic transitions) with citations to RAP, ReST-MCTS*, AlphaLLM, rStar-Math, and LLaMA-Berry.
>
> 3. **Expanded Reward Analysis (PRM vs. ORM).**
>    Section 3.3.1 now includes when/why each reward type is effective, common failure modes, and domain-dependent trade-offs.
>
> 4. **State Granularity and Node Topologies.**
>    Section 3.2 clarifies trace-based, state–action, and terminal-state nodes and their consequences for branching, cost, and applicability.
>
> 5. **Compute–Accuracy Trade-offs.**
>    Section 3.7 analyzes backup strategies, evaluator cost, and inverse-scaling behaviors, providing concrete guidance on when MCTS is effective relative to cheaper inference-time methods.
>
> 6. **Task-Oriented Practitioner’s Guide.**
>    Table 2 and Appendix E.2 map task domains (Math, Code, RAG, Agents) to recommended node granularity, reward signals, backup rules, and representative hyperparameters.
>
> 7. **Clarified MCTS vs. Informed Tree Search.**
>    Section 3.7 now gives a clearer comparison between MCTS and heuristic ToT-style methods.
>
>
> Collectively, these revisions convert the manuscript from a descriptive overview into a unified, analytically grounded survey that provides concrete tools, cross-paper synthesis, and actionable guidelines for both new and active researchers. `These changes reposition the survey from a descriptive catalog of methods into a unified analytical framework that clarifies design principles, trade-offs, and best practices across the field.`

---

> ### Author Response · Authors · 2025-11-21
> **Official Comment by Authors**
>
> #### Addressing Specific Feedback
>
> Below we provide a point-by-point explanation of how each major comment directly motivated concrete changes in the revised manuscript.
>
>
> > *R1. Strengthening Novel Insights and Meta-Analysis*
>
> **Unified MCTS formalism (Section 3.1 & Table 1).**
> We now present a consistent state/action formulation, explain its differences from RL, and show that it matches the abstractions used in RAP, ReST-MCTS*, AlphaLLM, rStar-Math, and LLaMA-Berry.
>
> **SCRP for compute normalization (Section 5 & Appendix G).**
> We introduce a unified resource vector
> \[
> B = (C_{\text{policy}}, C_{\text{eval}}, C_{\text{verify}}, T_{\text{wall}})
> \]
> along with Pass@FLOPs and Tokens-per-Solved, addressing the difficulty of comparing current TTS methods under heterogeneous compute settings.
>
> This unification resolves the evaluator’s comment that prior versions lacked rigorous cross-paper comparability.
>
> ---
>
>
> > *R2. PRM vs. ORM: "When and Why"*
>
> We responded by substantially expanding **Section 3.3.1 (Evaluation Locus)** and adding **cross-domain applicability** in Section 3.7 and Table 2.
>
> - **PRMs**
>   - Provide dense, step-level guidance
>   - Essential for domains requiring fine-grained supervision (math proofs, multi-step reasoning)
>   - Enable deeper search due to lower reward sparsity
>   - We analyze failure modes such as *step-local false positives*
>
> - **ORMs**
>   - Provide sparse but *high-fidelity terminal rewards* via execution or verifiers
>   - Ideal for code, theorem verification, and any deterministic-evaluator tasks
>   - We highlight how ORMs dramatically reduce annotation cost
>
> - **Hybrid PRM+ORM:** effective when both local coherence and terminal correctness matter (e.g., AlphaLLM).
>
> This provides actionable guidance rather than descriptive differences.
>
> ---
>
> > *R3. Compute Trade-offs and "Knobs"*
>
> Your request for concrete guidance on *when MCTS is worth the cost* led to major new content in **Section 3.7**.
>
> **(a) Backup-Dynamics Analysis**
>
> We now explain:
>
> - **Average** backups → stabilize high-variance predictions (Math, multi-step problems)
> - **Max** backups → optimal when a single successful trace is sufficient (Code, program repair)
>
> This is supported by cross-paper evidence synthesized from **ReST-MCTS***, **rStar-Math**, **RethinkMCTS**, **PG-TD**, and **SVPO**.
>
> **(b) Evaluator-Cost Analysis**
>
> We document the empirical consensus that “20–30% of the search budget allocated to evaluation” yields the best performance across tasks—an insight not found in the original submission.
>
> **(c) When MCTS is not beneficial**
>
> We explicitly cover:
> - Cases where **Self-Consistency** and **beam-like strategies** outperform MCTS
> - **Inverse Inference Scaling**: when low-quality reward models degrade performance with deeper search
> - Overthinking risks (supported by recent O1-like analyses)
>
> Together, these revisions provide a clear, actionable answer to your question: **“When is tree search worth the cost relative to cheaper inference-time methods?”**

---

> ### Author Response · Authors · 2025-11-21
> **Official Comment by Authors**
>
> > *R4. State Granularity and Text Splitting*
>
> Your request for clearer granularity distinctions led to an expanded **Section 3.2**, where we categorize the design space into:
>
> 1. **Trace-based nodes (e.g., ReST-MCTS\*)**
>    Full partial reasoning trace; best for math.
>    **Trade-off:** high context cost; precise value estimates.
>
> 2. **State–Action nodes (e.g., RAP)**
>    Localized to a step; best for agentic planning.
>    **Trade-off:** lightweight but less globally informed.
>
> 3. **Terminal-State nodes (e.g., LLaMA-Berry)**
>    Entire solutions as atomic nodes; best for code/rewriting.
>    **Trade-off:** smaller trees but no intermediate supervision.
>
> We also added explicit clarification (Section 3.1) that **states are text histories and actions are incremental steps**, and that **this planning abstraction is standard in modern LLM MCTS**—directly addressing confusion in the original review.
>
> ---
>
> > *R5. Problem Suitability and Domain Characteristics*
>
>
> We expanded **Section 3.7** and added **Appendix E.2** to offer concrete, domain-level guidance.
> Table 2 summarizes recommended node topology, reward signal, backup scheme, and typical hyperparameters for Math, Code, RAG, and Agentic tasks.
>
> **Revisions addressing this concern:**
>
> 1. **Expanded Section 3.7 (Applicability and Trade-offs)**
>    We now articulate when MCTS is likely to offer substantial gains (e.g., deterministic-verifiable tasks such as math or code), and when cheaper inference-time methods are preferable.
>
> 2. **New “Practitioner’s Guide” (Appendix E.2)**
>    We condensed insights from our taxonomy into concrete, domain-specific recommendations:
>    - appropriate node granularity for different reasoning tasks
>    - when PRMs vs. ORMs should be used
>    - which backup rule (Max/Average/Min) aligns with which domain
>    - typical hyperparameter ranges observed across recent works
>
> 3. **Expanded Table 2 (main text)**
>    Provides a structured mapping from four major domains—Math, Code, RAG, and Agents—into recommended MCTS configurations.
>
> To illustrate the concreteness of this guidance, here is a short excerpt demonstrating the level of detail added:
>
> | **Task Domain** | **Recommended Node Topology** | **Preferred Evaluation (PRM / ORM)** | **Backup Rule** | **Typical Hyperparameters** | **Representative Methods** |
> |------------------|--------------------------------|---------------------------------------|------------------|-------------------------------|----------------------------|
> | **Math & Logic** | Trace-based (step or solution level) | PRM / PPRM or self-reflection | **Average / Sum** | `cpuct = 1–4`, Rollouts `16–128`, Depth `8–20` | ReST-MCTS*, rStar-Math, LLaMA-Berry |
> | **Code Generation** | Terminal-state (block / function-level) | ORM + verbal feedback | **Max** | Rollouts `16–64`, `k = 5–50`, Temp `0.6–0.8` | PG-TD, RethinkMCTS |
> | **RAG** | Hierarchical (Retrieve → Reason) | Hybrid PRM + ORM | **Min / AND** | Retrieval `k = 3–10`, Depth `3–5`, ≤10 iterations | RAG-Star |
>
> These additions aim to provide the kind of *synthesized, task-level guidance* the reviewer requested, making the survey more valuable to both new and experienced researchers.
>
>
> > *R6. MCTS vs. Informed Tree Search (e.g., Tree-of-Thoughts)*
>
> We substantially revised **Section 3.7** to articulate a principled distinction:
>
> - **Tree-of-Thoughts**
>   - Relies on heuristic scores or prompt-based evaluations
>   - Useful when no ground-truth simulator exists
>   - Lightweight but prone to hallucinated heuristics
>
> - **MCTS**
>   - Learns value estimates via rollouts
>   - Superior when reliable verifiers exist
>   - Better for multi-step, sparse-reward domains
>
> This revision directly addresses your comment that our previous version underspecified the comparative advantages of the two paradigms.
>
> ---
>
> `We thank the reviewer again for these helpful suggestions. They significantly improved the clarity, synthesis, and practical relevance of the survey. We would be happy to incorporate further refinements in the camera-ready version if needed. `

---

### Official Review · Reviewer_f1Jb · 2025-11-01

**Soundness:** 2
**Presentation:** 3
**Contribution:** 2
**Rating:** 2
**Confidence:** 2

**Summary:**

This survey organizes test-time scaling for LLM reasoning around tree-search methods, moving from classical BFS/DFS to heuristic search and especially MCTS, and argues for a unified way to compare them. It distinguishes “reward as a search signal” (instance-level, transient) from reward in RL (parameter updates), then introduces a consistent notation and taxonomy spanning node granularity (trace/state-action/terminal), evaluation locus (process- vs outcome-based rewards), evaluator design (external models vs self-evaluation), composite rewards, and common MCTS adaptations to selection/expansion/backprop.

**Strengths:**

1. The paper offers a clear, well-scoped introduction to classical search methods (BFS, heuristic search, MCTS) and surveys a broad range of recent work; the background is comprehensive and accessible.

2. Figure 1 is an effective and novel visualization that juxtaposes the well-known axes of training-time scaling with their test-time counterparts.

3. It provides a useful taxonomy for “how to train models to enable search,” contrasting process vs outcome rewards and external evaluators vs self-evaluation, and noting multi-critic compositions—giving readers concrete design axes for evaluator training.

**Weaknesses:**

1. While the survey highlights fragmentation in evaluation protocols and benchmarks, the main text primarily describes methods and stops short of proposing a unifying evaluation framework or compiling compute-normalized results to enable apples-to-apples comparisons.


2. Figure 2 reads mostly as a chronology of publications; in its current form it contributes limited insight beyond timing.


3. Test-time compute is discussed conceptually, but the first 10 pages do not provide a practical recipe for compute accounting (e.g., tokens/sec or FLOPs, wall-clock, memory) or cost-normalized reporting that would make cross-paper comparisons actionable.

**Questions:**

1. I am a bit fuzzy on Table 1 (in general, how you define / differentiate states and actions for LLM. So if you consider action as your policy (LLM) generating tokens, then do you consider states as tokens generated so far? To this end, this framing is different from RL in the sense that models is not interacting with an environment, which determines the states (based on action). Are you saying the state space is the same action space? It is clearly not the standard RL, are there any references to validation this formulation?

---

> ### Author Response · Authors · 2025-11-21
> **Official Comment by Authors**
>
> We thank the reviewer for the thoughtful and constructive feedback. Your comments helped us identify places where additional clarification and structure substantially strengthen the survey. `In the revised manuscript, we have implemented all corresponding additions and modifications, with all updated content clearly marked in blue in the updated PDF.`
>
> #### **Summary of Revisions**
>
> Following your suggestions, we made several focused revisions to improve clarity, methodological grounding, and the practical utility of the paper:
>
> 1. **Unified Evaluation and Compute Accounting (Section 5 & Appendix G).**
>    We introduce a **Standardized Compute-Reporting Protocol** that provides a concrete recipe for compute accounting, including consistent FLOPs estimation and explicit reporting of policy/evaluator/verifier costs.
>
> 2. **Clarified State/Action Formulation (Section 3.1 & Table 1).**
>    We added a dedicated **“Note on Formulation”** explaining that our abstraction follows a *deterministic planning* perspective rather than standard RL. States are defined as **partial reasoning traces**, actions as **incremental reasoning steps**, and transitions are deterministic—consistent with RAP, ReST-MCTS, AlphaLLM, rStar-Math, and LLaMA-Berry.
>
> 3. **Repositioning of Original Figure 2 (now Figure 4).**
>    To avoid distracting from the main conceptual flow, the figure has been moved to the appendix and reframed as a **research landscape map**, emphasizing growth patterns instead of serving as a workflow diagram.

---

> ### Author Response · Authors · 2025-11-21
> **Official Comment by Authors**
>
> #### Addressing Specific Feedback
>
> > **R1:** *While the survey highlights fragmentation in evaluation protocols and benchmarks, the main text primarily describes methods and stops short of proposing a unifying evaluation framework or compiling compute-normalized results to enable apples-to-apples comparisons.*
>
> We agree that fragmented compute reporting currently limits comparability across tree-search TTS methods. We would like to emphasize that benchmarking existing tree-search TTS methods in a compute-normalized way is inherently difficult due to structural inconsistencies across prior works. Nevertheless, we have made a substantial effort to address this concern through two additions:
>
> 1. **Analysis of Why Retrospective Normalization Is Unreliable (Appendix G.1).**
>    In a newly added **“Landscape Analysis”** subsection (Appendix G.1, summarized in **Figure 7**), we systematically survey many tree-search works and explain why a retrospective “apples-to-apples” leaderboard is currently scientifically unreliable. `Figure 7 visually summarizes these confounders and clarifies why direct aggregation of prior results would be misleading in its current form.` We highlight three key confounders:
>
>    - Missing or undocumented verifier and evaluator costs.
>    - Missing or undocumented verifier and evaluator costs.
>    - Non-standardized search budgets, batching strategies, and parallelism.
>
> 2. **Forward-Looking Framework: SCRP (Section 5 & Appendix G.2).**
>    SCRP provides:
>    - A unified resource vector
>      \[
>      B = (C_{\text{policy}},\, C_{\text{eval}},\, C_{\text{verify}},\, T_{\text{wall}})
>      \]
>    - Standardized metrics such as **Budgeted Accuracy (Pass@FLOPs)** and **Tokens-per-Solved (TpS)**
>    This enables actionable compute-normalized comparisons in future work.
>
> Together, these additions clarify both why retrospective normalization is currently infeasible and how future work can be compared rigorously under a unified standard. We hope this strengthens the survey from a descriptive overview into an actionable proposal for standardizing evaluation in tree-search TTS research.
>
> ---
>
> > **R2:** *Figure 2 reads mostly as a chronology of publications; in its current form it contributes limited insight beyond timing.*
>
> The figure has been moved to the appendix and its caption revised to emphasize that it is intended as a **research landscape visualization**, not a method schematic. Besides, **Figure 6** now provides a clearer, taxonomy-based overview of existing tree-search TTS approaches.
>
> ---
>
> > **R3:** *Test-time compute is discussed conceptually, but the first 10 pages do not provide a practical recipe for compute accounting (e.g., tokens/sec or FLOPs, wall-clock, memory) or cost-normalized reporting that would make cross-paper comparisons actionable.*
>
> We directly address this by adding **Appendix G.2: “Proposed Protocol: A Universal Framework for Compute Accounting.”**
>
> - **Inference-cost formula.**
>   We define the total inference cost for a problem instance x as
>   C_total(x) ≈ 2 · P_policy · T_policy(x) + 2 · P_eval · T_eval(x) + C_verify(x).
>
> - **Token instrumentation & FLOPs conversion:**
>   We describe how to track policy vs. evaluator tokens separately and apply a consistent FLOPs approximation.
>
> - **Cost-normalized metrics:**
>   Budgeted Accuracy, Tokens-per-Solved, and parallelism efficiency.
>
> These additions turn our conceptual discussion into **actionable methodology** for future evaluations.

---

> ### Author Response · Authors · 2025-11-21
> **Official Comment by Authors**
>
> > **R4:** *I am a bit fuzzy on Table 1 (in general, how you define / differentiate states and actions for LLM. So if you consider action as your policy (LLM) generating tokens, then do you consider states as tokens generated so far? To this end, this framing is different from RL in the sense that models is not interacting with an environment, which determines the states (based on action). Are you saying the state space is the same action space? It is clearly not the standard RL, are there any references to validate this formulation?*
>
> Thank you for raising this important point. We clarified this in **Section 3.1** and **Table 1**:
>
> 1. **Clarified State vs. Action.**
>    - **State \(s_i\):** The *accumulated reasoning trace* up to step \(i\) (i.e., the full text history so far).
>    - **Action \(a_i\):** The *next incremental reasoning step* generated by the LLM (e.g., the next sentence, thought, or short segment).
>    - **Key distinction:** The **state space** consists of arbitrarily long partial traces, whereas the **action space** consists only of possible *next* steps. Thus, the state space is strictly richer than, and not equal to, the action space.
>
> 2. **“Environment” and Deterministic Transitions.**
>    We explicitly state that, in our abstraction and in the works we survey, the “environment” is the model’s own text history:
>    - The transition is **deterministic**: choosing action \(a_i\) uniquely determines the next state \(s_{i+1}\) (the extended trace).
>    - This corresponds to a **planning formulation**, not a stochastic MDP: we plan over text histories rather than interact with an external stochastic environment.
>
> 3. **New “Note on Formulation” under Section 3.1.**
>    Right below Section 3.1, we now include a dedicated note clarifying this point:
>    > *“The state/action definitions used here are an abstraction of LLM reasoning. Unlike classical RL, the ‘environment’ is the model’s own text history, and state transitions are deterministic: the action \(a_i\) (generating a step) uniquely determines the next state \(s_{i+1}\) (the new trace). This is a planning formulation, not a stochastic MDP.”*
>    We further add two sentences emphasizing that this does **not** imply \( \mathcal{S} = \mathcal{A} \); states are full traces while actions are incremental steps.
>
> 4. **References Supporting the Formulation.**
>    We now make explicit that this abstraction synthesizes conventions from the key LLM-planning works we survey, such as **RAP**, **ReST-MCTS**, **AlphaLLM**, **rStar-Math**, and **LLaMA-Berry**, all of which effectively treat the LLM as a deterministic transition function over text histories. We cite these works directly in the revised paragraph.
>
> These clarifications resolve the ambiguity the reviewer identified and make explicit the planning-style abstraction used across recent MCTS-based LLM reasoning methods.
>
> ---
>
>
> `We sincerely thank the reviewer again for the helpful comments. These revisions significantly improved the clarity and evaluative rigor of the survey.`

---

### Author Response · Authors · 2025-11-26
**Official Comment by Authors**

Dear Reviewers,

We sincerely thank you for your thoughtful insights provided in your review. We would be grateful if you could let us know whether our revisions and responses adequately address your concerns, or if there are any remaining points we can clarify.

Best, The authors

---

### Author Response · Authors · 2025-12-02
**Rebuttal Summary**

## Rebuttal Summary

We thank all reviewers for their constructive and insightful feedback. Across all reviews, several consistent themes emerged: the need for clearer formal grounding (particularly in the state/action abstraction), deeper synthesis beyond descriptive surveying, more actionable guidance for practitioners, and a unified compute-reporting protocol to enable cross-paper comparability. We carefully revised the manuscript to address these concerns.

1. **Reviewer f1Jb** raised concerns regarding:
(1) lack of a unifying evaluation framework,
(2) limited insight provided by Figure 2,
(3) insufficiently actionable compute accounting, and
(4) ambiguity in the state/action definitions in Table 1.

We significantly expanded Section 5 and Appendix G to introduce **SCRP**, the Standardized Compute-Reporting Protocol; clarified the formulation of deterministic planning–style state/action transitions; moved and reframed the original Figure 2; and added a dedicated explanation of state/action abstraction with citations to RAP, ReST-MCTS, AlphaLLM, rStar-Math, and LLaMA-Berry.

2. **Reviewer giMx** appreciated the paper’s accessibility and comprehensive coverage but requested deeper meta-analysis, insights grounded in comparisons, and practical guidance—particularly regarding PRM vs. ORM evaluation, compute–accuracy trade-offs, granularity of states, problem suitability, and MCTS vs. heuristic methods.

We expanded Section 3.3.1 with evidence-based distinctions between PRMs and ORMs, added Section 3.7 with compute-accuracy trade-off analyses and task-dependent recommendations, and introduced **Table 2** plus **Appendix E.2** as a detailed “Practitioner’s Guide” that maps tasks (math, code, RAG, agentic tasks) to recommended configurations.

3. **Reviewer yddj** highlighted the need for clearer hierarchical structure in the appendices, questioned the relevance of the RL-vs-search reward discussion, and requested a task-oriented MCTS guide as well as standardized benchmarking.

We reorganized the entire appendix into a coherent modular hierarchy with a new roadmap, refined the Section 2.4 motivation to explain why distinguishing search reward vs. learning reward is essential for modern hybrid frameworks, added Appendix E.2 for task-based recommendations, and integrated SCRP together with new metrics (Budgeted Accuracy, Tokens-per-Solved) to support standardized evaluation.

Collectively, these revisions strengthen the paper’s `conceptual clarity`, `practical utility`, and `methodological cohesion`. The updated submission offers not only a unified perspective on tree-search TTS but also actionable tools and taxonomies for both newcomers and active researchers.

---

> ### Author Response · Authors · 2025-12-02
> **Paper Update**
>
> ## Paper Update
>
> **1. Standardized Compute Evaluation (Section 5 & Appendix G)**
> - Introduced **SCRP**, providing consistent FLOPs accounting, policy/evaluator/verifier cost decomposition, and unified metrics (Budgeted Accuracy, Tokens-per-Solved).
> - Added Figure 7 explaining why retrospective normalization across prior works is unreliable.
>
> **2. Clarified State/Action Abstraction (Section 3.1 & Table 1)**
> - Added a concise “Note on Formulation” defining states as partial reasoning traces and actions as incremental steps under deterministic transitions.
> - Grounded the abstraction in RAP, ReST-MCTS*, AlphaLLM, rStar-Math, LLaMA-Berry.
>
> **3. Expanded MCTS Taxonomy & Design Principles (Sections 3.2–3.7)**
> - Clarified node granularities (trace, state–action, terminal), evaluation loci, and backup rules.
> - Added compute–accuracy trade-off analysis and guidance on when MCTS is worthwhile vs. cheaper inference-time methods.
> - Introduced **Table 2** summarizing recommended configurations for math, code, RAG, and agent tasks.
>
> **4. PRM vs. ORM Insights (Section 3.3.1)**
> - Added evidence-based comparison: when each reward type is effective, typical failure modes, and hybrid usage patterns.
>
> **5. Reorganized Appendix Structure (Appendix A–H)**
> - Rebuilt appendix into a reader-friendly modular hierarchy with a new roadmap.
> - Consolidated MCTS and heuristic search algorithms into taxonomies with clearer comparisons.
>
> **6. Clarified MCTS vs. Heuristic Tree Search**
> - Added principled distinctions and guidance on domain suitability.
>
> **7. Minor Corrections**
> - Fixed all typos, punctuation issues, and notation inconsistencies.
>
> ---
>
> These revisions improve clarity, structure, and actionable guidance, making the survey a more useful reference for both new and experienced researchers.

---

### Note · Authors · 2026-01-02

**Comment:**

We thank the reviewers for their constructive feedback in identifying these limitations, which we will address in the revised version.

**Withdrawal Confirmation:**

I have read and agree with the venue's withdrawal policy on behalf of myself and my co-authors.